# Gut opportunistic pathogens contribute to high-altitude pulmonary edema by elevating lysophosphatidylcholines and inducing inflammation

Xianduo Sun,[1] Gaosheng Hu,[1] Yuting Li,[2] Wenjing Li,[1] Yong Wang,[1] Hui Yan,[1] Guoqing Long,[1] Long Zhao,[1] Anhua Wang,[1] Jingming Jia[1]

**ABSTRACT**  Gut microbiota have been found to promote hypoxia-induced intestinal injury. However, the role of gut microbiota in high-altitude pulmonary edema (HAPE), the preventive effect of synbiotic on HAPE, and the mechanisms by which they might work remain unknown. In this study, we aimed to investigate the role of gut microbiota in the pathogenesis of HAPE and to explore the underlying mechanisms involved. We performed a fecal microbiome analysis and found a significant decrease in intestinal *Klebsiella* and *Escherichia-Shigella*, along with a notable increase in intestinal *Bifidobacterium* and *Lactobacillus*, as volunteers recovered from acute mountain sickness (AMS). Gavage colonization with *Klebsiella pneumoniae* and *Escherichia coli* induced plasma inflammation, increased plasma lysophosphatidylcholine (LPC) levels, and contributed to HAPE in rats at a simulated altitude of 6,500 m. Conversely, a synbiotic containing *Bifidobacterium*, *Lactiplantibacillus*, fructooligosaccharides, and isomaltose-oligosaccharides significantly reduced the severity of HAPE. Cellular experiments and molecular dynamics simulations revealed that LPCs can cause damage and permeability to human pulmonary microvascular endothelial cell (HPMEC) and human pulmonary alveolar epithelial cell (HPAEpiC) monolayers under hypoxic conditions by disrupting cell membrane integrity. In addition, tail vein injection of LPCs promoted HAPE in mice at a simulated altitude of 6,500 m. In conclusion, this study describes a gut microbiota-LPCs/inflammation-HAPE axis, an important new insight into HAPE that will help open avenues for prevention and treatment approaches.

**IMPORTANCE**  The role of the gut microbiota in high-altitude pulmonary edema (HAPE) is currently unknown. This study found that intestinal *Klebsiella pneumoniae* and *Escherichia coli* contribute to HAPE by inducing inflammation and increasing lysophosphatidylcholine (LPC) levels under hypoxic conditions. Conversely, a synbiotic containing *Bifidobacterium*, *Lactiplantibacillus*, fructooligosaccharides, and isomaltose-oligosaccharides significantly reduced the severity of HAPE. Further investigation revealed that LPCs can cause damage and permeability to human pulmonary microvascular endothelial cell (HPMEC) and human pulmonary alveolar epithelial cell (HPAEpiC) monolayers under hypoxic conditions by disrupting cell membrane integrity. These findings contribute to the understanding of the pathogenesis of HAPE and will benefit populations living at high altitude or traveling from low to high altitude.

**KEYWORDS**  high-altitude pulmonary edema, gut microbiota, lysophosphatidylcholines, inflammation

**Peer Reviewer** Jinshan Suo, Huashan Hospital, Fudan University, Shanghai, China

Address correspondence to Jingming Jia, jiajingming@syphu.edu.cn, or Anhua Wang, sywanganhua@163.com.

The authors declare no conflict of interest.

High-altitude pulmonary edema (HAPE) is a non-cardiogenic pulmonary edema whose symptoms include cough, shortness of breath, reduced exercise tolerance,

and the presence of frothy pink sputum with rapid breathing (1). In places where medical care is limited, the mortality rate of this illness can be as high as 40% (2). Autopsies of people who have died from HAPE show damage to the lung walls and protein-rich fluid and bleeding in the alveoli (3). Traditionally, an effective method of preventing HAPE is gradual ascent, and an effective method of treating HAPE includes descent and supplemental oxygen (4). Although nifedipine, salmeterol, tadalafil, dexamethasone, and acetazolamide are recommended for the prevention or treatment of HAPE, they may result in side effects (4, 5). The exact pathogenesis of HAPE is not fully understood. However, there are three accepted pathways: the "classical" hydrostatic hypothesis, the epithelial hypothesis involving impaired alveolar reabsorption, and the endothelial/inflammatory hypothesis (6). Unfortunately, the current understanding of how hypoxia leads to increased endothelial cell permeability and triggers inflammation is still limited (6).

More than 100–1,000 microbial species reside in the human gut (7) and are associated with various human diseases (8). Although gut microbiota have been found to promote hypoxia-induced intestinal injury (9) and are an indispensable factor in causing intestinal barrier damage in mice exposed to high altitude (10), their underlying role and mechanism in HAPE have received little attention. Research suggests that exposure to high altitude can significantly alter the gut microbiota (11, 12), which is characterized by an increase in anaerobic bacteria (13). The relative abundances of *Prevotella*, *Clostridiales*, *Clostridium*, *Lachnospiraceae,* and *Pseudobutyrivibrio* are higher in highlanders than in lowlanders. These taxa are known to produce short-chain fatty acids (SCFAs), which are not only an important source of energy but can also mediate host blood pressure (14). In contrast, the relative abundances of opportunistic pathogens, such as *Klebsiella* and *Escherichia coli*, are significantly reduced in the guts of highlanders, but their role in HAPE or other high-altitude illnesses has not been further investigated (15, 16). In addition, existing studies have mainly focused on changes in the gut microbiota between low- and high-altitude populations, while changes in the gut microbiota during HAPE or acute mountain sickness (AMS) have received little attention.

Studies have shown that disruption of lipid metabolism can lead to various diseases (17, 18). For example, high levels of oxidized phospholipids can promote inflammation and damage the endothelial cell barrier, leading to atherosclerosis (19). The gut microbiota are known to play an important role in regulating lipid metabolism through various products such as SCFAs, bile salt hydrolase, lipid metabolism-related enzymes, and others, thus influencing the development of lipid-related diseases (20). However, the link between gut microbiota-associated lipid metabolism and HAPE remains unclear.

Here, we tested the hypothesis that the gut microbiota contribute to the pathogenesis of HAPE. To this end, we performed microbiome analysis on 90 fecal samples from volunteers at different altitudes and severity of AMS to identify the beneficial and harmful microbiota for HAPE. The differential gut microbiota were transplanted into rats to determine their role in HAPE, and potential mechanisms were explored by detecting levels of inflammatory factors, transcriptome, and lipid metabolome. The roles and mechanisms of differential metabolites in HAPE were investigated using cell and mouse experiments and molecular dynamics simulations. Our results support the hypothesis that gut opportunistic pathogens contribute to the severity of HAPE by elevating lysophosphatidylcholines (LPCs) and inducing inflammation under hypoxic conditions.

## MATERIALS AND METHODS

### Study design and volunteers

A total of 51 volunteers participated in the study, consisting of 30 individuals residing at low altitude (12 females and 18 males) and 21 individuals living at high altitude (7 females and 14 males). All participants were over 20 years of age, had no gastrointestinal abnormalities or diseases, were not taking any medications, and had not undergone

a colonoscopy in the past 3 mo. Thirty low-altitude volunteers were taken to higher altitudes (Fig. S1), where the Lake Louise Acute Mountain Sickness (AMS) score (21) and fecal samples were collected daily. However, there were instances when stool samples from certain individuals could not be collected. Six volunteers did not experience AMS throughout the study period and were therefore excluded from the analysis. Samples labeled with "△" were designated as the low-altitude group (LG), those marked with "▽" were categorized as the high-altitude AMS (HA) group, and samples indicated by "◇" were classified as the AMS recovery (HR) group (Table S1). Fecal samples from the high-altitude volunteers constituted the high-altitude group (HG). All 90 fecal samples were stored at −80°C for 16S rRNA sequencing.

## Microbiome analysis

Total microbial DNA was extracted from fecal samples using the PowerSoil DNA Isolation Kit (MoBio Laboratories, Cat# 12888) according to the instructions, and the purity and quality of the total DNA were checked using the NanoDrop spectrophotometer (Thermo Scientific). The V3-4 hypervariable region was amplified with the primers 338F and 806R (22). For each sample, an eight-digit barcode sequence was added to the 5′ end of the forward and reverse primers. The PCR was carried out on a Mastercycler Gradient (Eppendorf, Germany) using 25 µL reaction volumes, containing 12.5 µL 2× Taq PCR MasterMix, 3 µL bovine serum albumin (BSA) (2 ng/µL), 1 µL forward primer (5 µM), 1 µL reverse primer (5 µM), 2 µL template DNA, and 5.5 µL ddH2O. Cycling parameters were 95°C for 5 min, followed by 28 cycles of 95°C for 45 s, 55°C for 50 s, and 72°C for 45 s with a final extension at 72°C for 10 min. The PCR products were purified using an Agencourt AMPure XP Kit (Beckman Coulter, Cat# A63881).

Deep sequencing was performed using the Illumina MiSeq platform. The raw data were de-multiplexed with an in-house Perl script, followed by quality filtering using fastp v0.19.6 (23) and merging with FLASH v1.2.11 (24) according to the following criteria: (i) reads were truncated at any position where the average quality score fell below 20 across a 50 bp sliding window, and truncated reads shorter than 50 bp were discarded; reads containing ambiguous characters were also excluded; (ii) only overlapping sequences longer than 10 bp were assembled based on their overlapping regions, with a maximum mismatch ratio of 0.2 in the overlap; reads that could not be assembled were discarded; (iii) samples were distinguished by barcodes and primers, with the sequence direction adjusted for precise barcode matching, allowing for a maximum of two nucleotide mismatches in primer matching. Qualified reads were clustered into operational taxonomic units (OTUs) at a 97% similarity threshold using the Uparse algorithm (25). The Ribosomal Database Project Classifier tool was utilized to classify all sequences into various taxonomic groups based on the Silva138 database (26).

Bioinformatic analysis of the gut microbiota was performed using the Majorbio Cloud platform (https://cloud.majorbio.com). Based on the OTU information, rarefaction curves and alpha diversity indices, including Chao1 richness and Shannon index, were analyzed using Mothur v1.30.2 (27). The similarity among the microbial communities in different samples was assessed by partial least squares discriminant analysis (PLS-DA) using R v3.3.1. Additionally, Python v2.7 software was employed for LEfSe analysis to identify bacterial taxa with significant differences in relative abundance between different groups (Linear Discriminant Analysis [LDA] score > 2, *P* < 0.05).

## Isolation, identification, and preparation of strains

Fresh fecal samples collected from volunteers were utilized for the isolation of target strains. Two strains, designated as KP and EC, were isolated from the HA group using Luria-Bertani (LB) solid medium, which contained 10 g/L peptone, 10 g/L NaCl, 5 g/L yeast extract, and 15 g/L agar, with a pH of 7.0. Similarly, two strains designated as BP and BL were isolated from the HR group using BBL solid medium, containing 15 g/L peptone, 2 g/L yeast extract, 20 g/L glucose, 0.5 g/L soluble starch, 5 g/L NaCl, 0.5 g/L L-cysteine, 400 mL/L tomato extract, 80 mL/L liver extract, and 20 g/L agar, with a pH

of 7.0. Additionally, one strain designated as LP was isolated from the HG group using De Man–Rogosa–Sharpe (MRS) solid medium, which comprised 10 g/L peptone, 5 g/L beef extract, 4 g/L yeast powder, 20 g/L glucose, 2 g/L $K_2HPO_4$, 2 g/L $(NH_4)_2HC_6H_5O_7$, 5 g/L $CH_3COONa$, 0.2 g/L $MgSO4·7H_2O$, 0.04 g/L $MnSO_4·H_2O$, and 18 g/L agar, with a pH of 6.8. The 16S rRNA gene sequences were amplified using universal primer pairs 27F and 1492R, followed by forward and reverse sequencing. The sequences were initially compared to the NCBI database, and two sequences with 100% similarity were identified as the same species. For sequences with less than 100% similarity, comparisons were made against the EzBioCloud database, and the sequences with the highest similarity were subsequently downloaded. Finally, the sequences were aligned, and an Neighbor-Joining (NJ) phylogenetic tree was constructed using the MEGA software.

LB broth was used to cultivate *Klebsiella pneumoniae* (*K. pneumoniae*) and *Escherichia coli* (*E. coli*) at 37°C for 12 h. BBL broth was employed for the anaerobic cultivation of *Bifidobacterium longum* (*B. longum*) and *Bifidobacterium pseudocatenulatum* (*B. pseudocatenulatum*) for 24 h at 37°C, while MRS broth was utilized to culture *Lactobacillus plantarum* (*L. plantarum*) for 12 h at the same temperature. Following cultivation, bacterial pellets were obtained through centrifugation, resuspended, and washed twice with sterile phosphate-buffered saline (PBS). The concentration of the strains was adjusted with sterile PBS to achieve a pathogenic suspension containing approximately 5 × $10^9$ CFU/mL of both *K. pneumoniae* and *E. coli*. Additionally, a probiotic suspension containing approximately 5 × $10^9$ CFU/mL of *L. plantarum*, 2.5 × $10^9$ CFU/mL of *B. longum*, 2.5 × $10^9$ CFU/mL of *B. pseudocatenulatum* (28), and 20 mg/mL each of fructooligosaccharides (29) and isomaltose-oligosaccharides was prepared, which was used as a synbiotic.

## Animal experiment

Male Sprague-Dawley rats and C57BL/6 mice were purchased from Liaoning Changsheng Biotechnology and were kept in a controlled environment with a temperature of 20 ± 5°C at low altitude (<500 m), a 12 h light/dark cycle, and unlimited access to food and water. Prior to the experiment, all animals were given time to acclimatize.

To evaluate the effect of target strains on HAPE, the rats were randomly divided into five groups: normoxia control (NC) group, hypoxia control (HC) group, normoxia + *E. coli* and *K. pneumoniae* (NKE) group, hypoxia + *E. coli* and *K. pneumoniae* (HKE) group, and hypoxia + *E. coli* and *K. pneumoniae* + synbiotics (HKE+S) group (Fig. S2). Each group comprised of eight rats. For the NC group, rats were administered sterile water orally twice a day for 3 d, followed by sterile PBS orally twice a day for 6 d. Throughout the experiment, the rats in the NC group were kept under normoxic conditions. In the HC group, rats received sterile water orally twice a day for 3 d, followed by sterile PBS orally twice a day for 3 d. Next, the rats were placed in a hypobaric chamber (Yuyan Instruments) designed to simulate an altitude of 6,500 m and were given sterile PBS orally twice a day for 3 d. In the NKE group, rats were treated with an antibiotic mixture (30) orally, administered twice a day for 3 d. Following this, a pathogenic suspension containing *K. pneumoniae* and *E. coli* was given orally, also twice a day for 6 d. Throughout the experiment, the rats in the NKE group were kept under normoxic conditions. In the HKE group, rats were treated with the same antibiotic mixture orally, administered twice a day for 3 d. Following this, the pathogenic suspension was given orally, also twice a day for 3 d. Afterward, the rats were again given the pathogenic suspension twice daily and were placed in a hypobaric chamber simulating an altitude of 6,500 meters for 3 d. For the HKE+S group, rats received the same antibiotic mixture orally, administered twice daily for 3 d, followed by the pathogenic suspension orally twice daily for another 3 d. Subsequently, the rats were given a synbiotic orally twice a day for 3 d. Finally, they were given the synbiotics orally twice daily and were placed in a hypobaric chamber at the same simulated altitude for 3 d. Each rat received a gavage dose of 10 mL/kg body weight at 8 AM and 8 PM.

To evaluate the effect of LPCs on HAPE, the mice were randomly divided into four groups: normoxia control (NC), normoxia + LPC (N+LPC), hypoxia control (HC), and hypoxia + LPC (H+LPC), with eight mice in each group. After 10 d of adaptation, the mice in the NC and HC groups received daily tail vein injections of PBS, while those in the N+LPC and H+LPC groups were administered LPCs at a dosage of 10 mg/kg body weight daily. Concurrently, the mice in the HC and H+LPC groups were placed in a hypobaric chamber simulating 6,500 m altitude for 3 d, whereas the mice in the NC and N+LPC groups were maintained under normoxic conditions for the same duration. Each mouse received an injection volume of 100 µL.

## The optimal prebiotics of *B. longum* and *L. plantarum* analysis

*K. pneumoniae* and *E. coli* strains in the logarithmic growth phase were diluted to an optical density ($OD_{600}$) of 0.4. Then, the diluted broth was added to the basal salt broth at a volume fraction of 1%, which contained $(NH_4)_2SO_4$ at 2 g/L, $MgSO_4 \cdot 7H_2O$ at 0.2 g/L, $Na_2HPO_4 \cdot H_2O$ at 0.5 g/L, $CaCl_2 \cdot 2H_2O$ at 0.1 g/L, and $K_2HPO_4$ at 0.5 g/L, with the pH adjusted to 7.2. Each basal salt broth was supplemented with one of the following: 10 g/L of inulin (shanghaiyuanye, cat# S11143), soybean oligosaccharide (shanghaiyuanye, cat# S25625), fructooligosaccharide (shanghaiyuanye, cat# S11133), galactooligosaccharide (shanghaiyuanye, cat# S11138), konjac glucomannan (shanghaiyuanye, cat# S30903), isomaltooligosaccharide (shanghaiyuanye, cat# S11134), glucose (shanghaiyuanye, cat# A10014), or no carbon source. The basal salt broth was then incubated at 37°C for 13 h, and the absorbance value was measured. On the other hand, *L. plantarum* and *B. longum* strains in the logarithmic growth phase were diluted to an $OD_{600}$ value of 0.4. Then, the diluted broth was added to MRS broth at a volume fraction of 1%, supplemented with one of the following carbon sources: 10 g/L inulin, soybean oligosaccharide, fructo-ligosaccharide, galactooligosaccharide, konjac glucomannan, isomaltooligosaccharide, glucose, or no carbon source. The MRS broth supplemented with *L. plantarum* or *B. longum* strains was incubated at 37°C for 9 or 29 h, respectively. Three replicates were conducted for each group, with each carbon source constituting a separate group. At the end of the cultivation, the absorbance value of $OD_{600}$ was measured. The difference in $OD_{600}$ values between each group with different carbon sources and the group without carbon sources was calculated to investigate the effect of different prebiotics on probiotic growth.

## The inhibitory effect of synbiotics on the growth of *E. coli* and *K. pneumoniae*

The modified MRS broth containing fructooligosaccharides (5 g/L) and isomaltooligosac-charides (5 g/L) as carbon sources was used to culture *K. pneumoniae*, *E. coli*, *L. plantarum*, and *B. longum*, respectively. The cultures were grown to the logarithmic growth phase and then diluted to an $OD_{600}$ of 0.40. For the KE group, 1 mL of *K. pneumoniae* and 1 mL of *E. coli* diluted broths were mixed with 2 mL of fresh modified MRS broth. This mixed broth was then added to the fresh modified MRS broth at a volume fraction of 1%, followed by anaerobic incubation at 37°C for 12 h. For the KE+BL group, 1 mL of *K. pneumoniae*, 1 mL of *E. coli*, 1 mL of *L. plantarum,* and 1 mL of *B. longum* diluted broths were mixed. This mixed broth was then added to the fresh modified MRS broth at a volume fraction of 1%, followed by anaerobic incubation at 37°C for 12 h. Three replicates were performed for each group. After incubation, 100 µL of each culture broth was plated and incubated on a basal salt medium containing 1% glucose for 72 h at 37°C. Colony counts were calculated to assess the effect of synbiotics on the growth of *E. coli* and *K. pneumoniae in vitro*.

## Lung water content

At the end of treatment for animals in each group, the wet weight of the lower lobe of the right lung tissue was measured immediately using a precision electronic balance. Lung samples were then dried at 90°C for 72 h to obtain a constant dry weight. Lung

water content was determined using Elliot's formula: lung water content (%) = (wet weight − dry weight) / wet weight × 100% (31).

## Histological analysis

A portion of the left lung was removed and preserved in 4% paraformaldehyde (Solarbio, Cat# P1110) at room temperature for 24 h. This sample was then embedded in paraffin and sliced into 3–4 µm sections. The sections were stained with hematoxylin-eosin (H&E) and examined under a light microscope to analyze the pathological changes in the rat lung tissues. Lung tissue pathology scores were evaluated as previously described (32).

## Measurement of plasma cytokines

At the end of treatment for rats in each group, plasma samples were collected for cytokine detection. IL-6 (Cloud-Clone, Cat# LMA079Ra), TNF-α (Cloud-Clone, Cat# LMA133Ra), IL-1β (Cloud-Clone, Cat# LMA563Ra), IL-17 (Cloud-Clone, Cat# LMA063Ra), and MCP-1 (Cloud-Clone, Cat# LMA087Ra) were assessed using Luminex200, all according to the manufacturer's protocols.

## Transcriptome analysis

RNA was extracted from the left lung of rats using TRIzol reagent (Majorbio), followed by quality evaluation with 5300 Bioanalyser (Agilent) and NanoDrop-2000 (NanoDrop Technologies). The sequencing library was then constructed with the high-quality RNA samples (OD260/280 = 1.8~2.2, OD260/230 ≥ 2.0, RIN ≥ 6.5, 28S:18S ≥ 1.0, >1 µg). RNA purification, reverse transcription, library construction, and sequencing were performed at Shanghai Majorbio Bio-pharm Biotechnology Co., Ltd. (Shanghai, China). The details of the experiment were as previously described (33).

After trimming and performing quality control on the raw paired-end reads with fastp (23) using default parameters, the clean reads were aligned to the reference genome in orientation mode using HISAT2 (34). A reference-based approach was used to assemble the mapped reads from each sample using StringTie (35).

The transcripts per million reads (TPM) method was applied to calculate transcript expression levels. Gene abundance was quantified with RNA-Seq by Expectation-Maximization (RSEM) (36). Essentially, DESeq2 (37) or DEGseq (38) was used to carry out differential expression analysis. Genes with $|log2FC| \geqq 1$ and false discovery rate (FDR) ≤ 0.05 (DESeq2) were defined as significantly differentially expressed genes (DEGs). In addition, Kyoto Encyclopedia of Genes and Genomes (KEGG) functional enrichment analyses were performed to identify which DEGs were significantly enriched in pathways with a $P$ value ≤ 0.05. KOBAS (39) was utilized for conducting KEGG pathway analyses.

## Lipidomics analysis

A total of 200 µL of plasma sample was accurately pipetted into an eppendorf (EP) tube. Subsequently, 80 µL of methanol (Thermo, Cat# 01162578/A452-4) and 400 µL of MTBE (Adamas, Cat# 01030358-28130F) were added. The mixture was vortexed and mixed for 30 s, followed by extraction through ultrasonication at 5℃ and 40 KHz for 30 min. The samples were then allowed to stand at −20℃ for 30 min and centrifuged at 13,000 × $g$ for 15 min at 4℃. Following centrifugation, 350 µL of the supernatant was removed, dried in a vacuum concentrator, and redissolved by adding 100 µL of the reagent solution (isopropanol [IPA] [HONEYWELL, Cat# 01377055/AH323-4]:acetonitrile [ACN] [Anpel, Cat# CAEQ-4-003306-4000] in a 1:1 ratio). After vortex mixing for 30 s, the sample was sonicated at 40 KHz for 5 min in an ice-water bath. The extracted lipids were then spun in a benchtop centrifuge at 13,000 × $g$ for 15 min at 4℃. The clarified supernatant was transferred to sample vials, and 5 µL portions of each sample were injected into the ultra-high performance liquid chromatography–tandem mass spectrometry (UHPLC-MS/MS) system. Quality control (QC) samples were prepared by

mixing equal volumes of all samples, which were handled and tested in the same manner as the analytical samples. These QC samples were injected at regular intervals (every 5–15 samples) to monitor the stability of the analysis.

UHPLC-MS/MS analysis was conducted using a Thermo UHPLC-Q Exactive HF-X Vanquish Horizon system equipped with an Accucore C30 column (100 mm × 2.1 mm i.d., 2.6 µm; Thermo). The mobile phase A consisted of 10 mM ammonium acetate (Guoyao, Cat# 10001216) in ACN and water (Thermo, Cat# W6-4) (1:1, vol/vol) with 0.1% (vol/vol) formic acid (Anpel, Cat# CAEQ-4-014784-0500), while mobile phase B comprised 2 mM ammonium acetate in ACN:IPA:water (10:88:2, vol/vol/vol) with 0.02% (vol/vol) formic acid. The standard sample injection parameters included a volume of 5 µL, a flow rate of 0.4 mL/min, a column temperature of 40°C, and a chromatographic separation time of 20 min. The solvent gradient was as follows: a linear gradient from 35% to 60% B from 0 to 4 min, from 60% to 85% B from 4 to 12 min, from 85% to 100% B from 12 to 15 min, held at 100% B from 15 to 17 min, from 100% to 35% B from 17 to 18 min, and maintained at 35% B until the end of the separation.

Mass spectrometric data were acquired using a Thermo UHPLC-Q Exactive HF-X benchtop Orbitrap mass spectrometer, operating in both positive and negative ion modes. The optimal conditions included an auxiliary gas flow rate of 20 psi, an auxiliary gas heater temperature of 37°C, a sheath gas flow rate of 60 psi, an ion spray voltage of −3,000 V in negative mode and +3,000 V in positive mode, and a normalized collision energy of 15-30-45 V for MS/MS. Data were collected using Data Dependent Acquisition (DDA) mode, enabling detection across a mass range of 200–2,000 m/z.

Data pre-processing and annotation were performed following ultra-performance liquid chromatography–tandem mass spectrometry (UPLC-MS/MS) analyses. The raw data were processed using LipidSearch (Thermo, California, USA) for peak detection, alignment, and identification. The mass tolerance for both the precursor and fragment was set at 10 ppm, and the M-score was set at 2.0, utilizing A, B, C, and D grades for the identification quality filter. MS/MS fragments were used to identify lipids, resulting in a data matrix containing lipid class, retention time, mass-to-charge ratio (m/z) values, and peak intensity.

The online majorbio platform (cloud.majorbio.com) was used for data analysis. The lipidomic features detected in at least 80% of each sample set were retained. Following the filtering process, minimum metabolite values were imputed for specific samples with values below the lower limit of quantification. Each metabolite feature was then normalized by summation. The response intensity of the sample mass spectrum peaks was normalized using the sum normalization method to obtain the normalized data matrix. Variables with a relative standard deviation greater than 30% of the QC samples were excluded, and the final data matrix was obtained after log10 transformation for subsequent analysis.

Differential metabolite analysis was conducted after data pre-processing, employing analysis of variance (ANOVA) on the matrix file. The R package ropls (version 1.6.2) was used to conduct PLS-DA and orthogonal partial least squares discriminant analysis (OPLS-DA), with model stabilities assessed using seven-cycle interactive validation. Additionally, significantly different metabolites were defined using variable importance in projection (VIP) from the OPLS-DA model with a value greater than 1 and a $P$ value of the Student's $t$-test less than 0.05.

Metabolites that differed between groups were summarized and then assigned to their respective biochemical pathways using metabolic enrichment and pathway analysis based on the KEGG database searches. These metabolites were categorized according to the pathways they were associated with or the functions they performed. The Python library Scipy Stats was utilized to determine statistically significant enriched pathways using Fisher's exact test.

## Cell culture

Human pulmonary microvascular endothelial cells (HPMECs) and human pulmonary alveolar epithelial cells (HPAEpiCs) were purchased from Otwo Biotech, Inc. (Shenzhen, China) and cultured in a humidified incubator with 5% $CO_2$ and 95% air at 37℃. HPMECs were cultured in endothelial cell medium (ScienCell, Cat# 1001), while HPAEpiCs were cultured in Dulbecco's modified Eagle's medium (meilunbio, Cat# MA0212) supplemented with 10% fetal bovine serum (meilunbio, Cat# PWL001), 100 U/mL penicillin, and 100 mg/mL streptomycin (meilunbio, Cat# MA0110). Cultured cells up to passage 3 with approximately 80% confluence were digested with 0.25% trypsin (meilunbio, Cat# MA0233) for subsequent experiments.

## MTT assay

Cells were cultured in 96-well plates at a density of $1 \times 10^4$ cells per well until they reached 90% confluence. Subsequently, the cells were treated synchronously with a corresponding serum-free medium for 10 h. Following this treatment, the cells were exposed to the corresponding serum-free medium containing LPCs (Aladdin, Cat# L130339) at concentrations of 1.56, 6.25, 25, or 100 µg/mL (10 mg/mL LPC stock solution was prepared in sterile PBS) or L-α-phosphatidylethanolamines (PEs) (Aladdin, Cat# D130313) at concentrations of 1.56, 6.25, 25, or 100 µg/mL (10 mg/mL PE stock solution was dissolved in sterile 65% ethanol) in a hypoxic incubator chamber (Billups-Rothenberg) for 6 h. The concentrations of $CO_2$, $O_2$, and $N_2$ in the hypoxia incubator chamber were adjusted to 5%, 4%, and 91%, respectively (40). The blank control group was treated with the corresponding medium without any drugs (serving as the blank control for the LPC treatment group) or with the corresponding medium containing 0.65% ethanol (serving as the blank control for the PE treatment group). In a separate experiment, after the synchronous treatment, HPMECs and HPAEpiCs cells were co-treated with the corresponding serum-free medium containing LPCs at 100 µg/mL and one of the inhibitors: Z-VAD-FMK at 10 µM (Aladdin, Cat# Z408507), nifedipine at 100 µM (Aladdin, Cat# N409014), glutathione at 1 mM (Aladdin, Cat# G105426), chloroquine at 25 µM (Aladdin, Cat# C424619), necrostatin-1 at 10 µM (MCE, Cat# HY-15760), liproxstatin-1 at 200 µM (MCE, Cat# HY-12726), and ferrostatin-1 at 1 µM (MCE, Cat# HY-100579) in an atmosphere of 4% (vol/vol) oxygen for 6 h to investigate the roles of apoptosis, calcium channels, oxidative stress, autophagy, necrosis, and ferroptosis in LPC-induced cell death (41–43). The blank control group was treated with the corresponding inhibitor without LPCs, while the negative control group received dimethyl sulfoxide (DMSO) alone.

After the above treatment, the culture medium was removed, and the cells were washed twice with PBS. Each well was supplemented with 100 µL of the appropriate serum-free medium and 10 µL of MTT (Sigma-Aldrich, Cat# M2128) solution (5 mg/mL) and was incubated at 37℃ for 4 h. The medium was then removed, and 100 µL of DMSO (Aladdin, Cat# D103277) was added to dissolve the formazan crystals that formed. A multifunctional microplate reader (BioTek) was used to measure the absorbance at 490 nm for each well. The survival rate was determined using the following formula: Survival rate = (experimental group OD value – negative control group OD value) / (blank control group OD value – negative control group OD value) × 100%.

## Evans blue permeability test

HPMECs (200 µL) and HPAEpiCs ($1 \times 10^5$ cells/mL) were seeded onto the polyester membrane cell culture insert (Labselect, pore size: 1 µm) with corresponding medium, while 600 µL of corresponding medium was placed in the lower chamber of 24-well dishes. The cells were synchronously treated with serum-free medium for 10 h when they had grown to 90% confluence. Subsequently, the cells were cultured with serum-free medium containing different concentrations of LPCs (0, 1.56, 6.25, 25, and 100 µg/mL) in the hypoxia incubator chamber by adjusting to 5% (vol/vol) $CO_2$, 4% (vol/vol) $O_2$, and 91% (vol/vol) $N_2$ concentrations for 6 h. After this treatment, the medium in the

lower chamber was replaced with 0.5 mL of PBS, and the cell culture insert was filled with 0.2 mL of 0.25% Evans blue (Macklin, Cat# E808783). After incubation for 0.5 h, the solution in the lower compartment was assayed for absorbance at 450 nm using a multifunctional microplate reader (BioTek) (44).

## Lactate dehydrogenase (LDH) detection

The procedures for cell processing are detailed in the MTT assay section. After incubating the cells with LPCs, the culture supernatant was collected for further analysis. To assess cell membrane damage, we measured lactate dehydrogenase (LDH) release using an LDH assay kit (Nanjing Jiancheng, Cat# A020-2), following the manufacturer's instructions.

## Scanning electron microscope

HPMECs and HPAEpiCs were cultured until they reached 80% confluence and were subsequently treated with corresponding serum-free media containing LPCs at concentrations of 25 and 100 µg/mL, respectively. The cultures were maintained in a hypoxia incubator chamber with adjusted concentrations of 5% (vol/vol) $CO_2$, 4% (vol/vol) $O_2$, and 91% (vol/vol) $N_2$ for 6 h. Following this, the cells were collected and fixed with 2.5% glutaraldehyde (Solarbio, Cat# P1126) for 24 h. After removal of the fixative, the cells were soaked twice with PBS for 10 min each time. They were then fixed with 1% osmic acid for 1 h at 4°C and then soaked twice with PBS for 10 min each time. The cells were then dehydrated through a series of graded alcohols and were dried using the critical point drying programme on Autosamdri-815, Series A (Tousimis). Finally, the cells were conductivity treated using a vacuum spray method before being observed and photographed using a scanning electron microscope (Hitachi).

## Molecular dynamics simulation

In the simulation setup, LPC molecules were randomly distributed across a mammalian cell membrane at a distance of 1 to 5 nm. The system included representative components of mammalian cell membranes such as cholesterol (CHL1), sphingolipids (PSM), phosphatidylcholine (POPC), phosphatidylserine (POPS), and phosphatidylethanolamine (POPE) (45). In addition, LPC molecules were placed directly above the cell membrane, including solvent molecules (mainly water) and physiological concentrations of sodium ions and chloride ions (Fig. S3). The simulation box was set to a size of $10 \times 10 \times 22$ nm$^3$. Based on the required concentration of 100 µg/mL LPC with a molar mass of 519.3 g/mol, this corresponds to approximately 50 LPC molecules in a volume of 2,200 nm$^3$. At a concentration of 6.25 µg/mL LPC, there were approximately three molecules in the same volume.

Using the lipid amounts given in Table S2, the cell membrane was constructed using the CHARMM-GUI (46, 47) platform. Lipids and solvents were simulated using the CHARMM36 all-atom force field and the TIP3P water model, respectively. The initial unit cell dimensions were (Lx, Ly, Lz) = (10, 10, 22) nm. Starting from each initial structure, a 10 ns equilibration Molecular Dynamics (MD) simulation was performed, followed by a 100 ns simulation in the NPT ensemble at a pressure of P0 = 1 bar and a temperature of T = 300 K. Simulations were performed using the GROMACS software package (48, 49). The pressure and temperature of the system were kept at the desired values using the Berendsen coupling method with a coupling constant of 0.1 ps. The motion equations were integrated using the leapfrog algorithm with a time step of 2 fs. Electrostatic interactions were calculated using the Particle Mesh Ewald method with a cutoff radius of 1.4 nm. Van der Waals interactions were also calculated with a cutoff radius of 1.4 nm.

## Statistical analysis

Statistical analysis was performed using SPSS 19.0 software, and data were expressed as mean ± standard deviation (SD). For data with normal distribution, a two-tailed Student's *t*-test or one-way ANOVA was used. Statistical significance compared to control was determined using a two-tailed, unpaired Student's *t*-test. Comparisons between multiple groups were made using one-way ANOVA. For non-parametric distributions, the Mann-Whitney U test or the Kruskal-Wallis test was used for comparisons between groups, and the resulting *P* values were adjusted using the Bonferroni method.

## RESULTS

### Hypoxia alters the human gut microbiota

The Lake Louise AMS score (Table S1) showed that 43% of low-altitude volunteers experienced AMS on the first day at high altitude. Symptoms worsened in most volunteers when they reached 4,400 m over 3–4 d. However, the symptoms gradually abated, and all volunteers recovered by day 10 even at 4,100 m. To investigate changes in the gut microbiota under high-altitude exposure and their relationship to AMS, fecal samples from low-altitude volunteers were categorized into three groups: the low-altitude group (LG), the high-altitude AMS (HA) group, and the AMS recovery (HR) group. Additionally, fecal samples from high-altitude volunteers constituted the high-altitude group (HG). All samples were used for microbiome analysis. After data optimization, we obtained a total of 7,410,203 reads from the fecal samples, averaging 82,336 ± 59,224 reads per sample. We evaluated whether the sample size for each group was sufficient by using the Pan/Core species curve. The results indicated that the curves began to plateau as the number of samples increased, confirming that the sample size was adequate (Fig. 1A and B). Additionally, the rarefaction curves based on the Chao1 and Shannon indices demonstrated that as the number of sequences increased, the curves gradually leveled off, indicating sufficient sequencing depth (Fig. 1C and D). Subsequently, we assessed both alpha and beta diversity of the gut microbiota. We found no differences between groups in the Chao1 and Shannon indices (Fig. 1E and F). However, beta diversity, as measured by PLS-DA, revealed differences. There was a clear separation between the HA and LG groups, as well as between the LG and HG groups. The beta diversity was more similar between the HR and HG groups compared to the differences observed between the LG and HG groups (Fig. 1G). Furthermore, the microbial dysbiosis index (MDI) was significantly elevated in the HG group compared to the LG group, as well as in the HA group compared to the HR group (Fig. 1H and I).

A linear discriminant analysis effect size (LEfSe) was performed to identify statistically significant biomarkers at the genus and species levels. The study revealed that the HR group exhibited a significantly lower relative abundance of *Escherichia-Shigella*, *Clavibacter michiganensis*, *Lactococcus*, and *Klebsiella* compared to the HA group. Conversely, the HR group showed a significantly higher relative abundance of *Bifidobacterium*, *Bifidobacterium pseudocatenulatum*, *Anaerostipes caccae*, *Bacteroides coprophilus*, and *Lactobacillus* (Fig. 1K). Notably, the HG group had a significantly higher relative abundance of *Bifidobacterium pseudocatenulatum*, alongside a significantly lower relative abundance of *Escherichia-Shigella* and *Klebsiella* compared to the LG group (Fig. 1L). It is well established that the genera *Lactobacillus* and *Bifidobacterium* are commonly utilized as probiotics. Furthermore, an analysis of species composition indicated that the genera *Bifidobacterium*, *Escherichia-Shigella*, and *Klebsiella* were among the top 25 most abundant genera (Fig. 1J). Therefore, we focused our research on the genera *Lactobacillus*, *Bifidobacterium*, *Escherichia-Shigella*, and *Klebsiella*. We isolated and identified an intestinal strain of *K. pneumoniae* and an intestinal strain of *E. coli* from the HA group, an intestinal strain of *B. pseudocatenulatum* and an intestinal strain of *B. longum* from the HR group, and an intestinal strain of *Lactobacillus plantarum* from the HG group (Fig. S4).

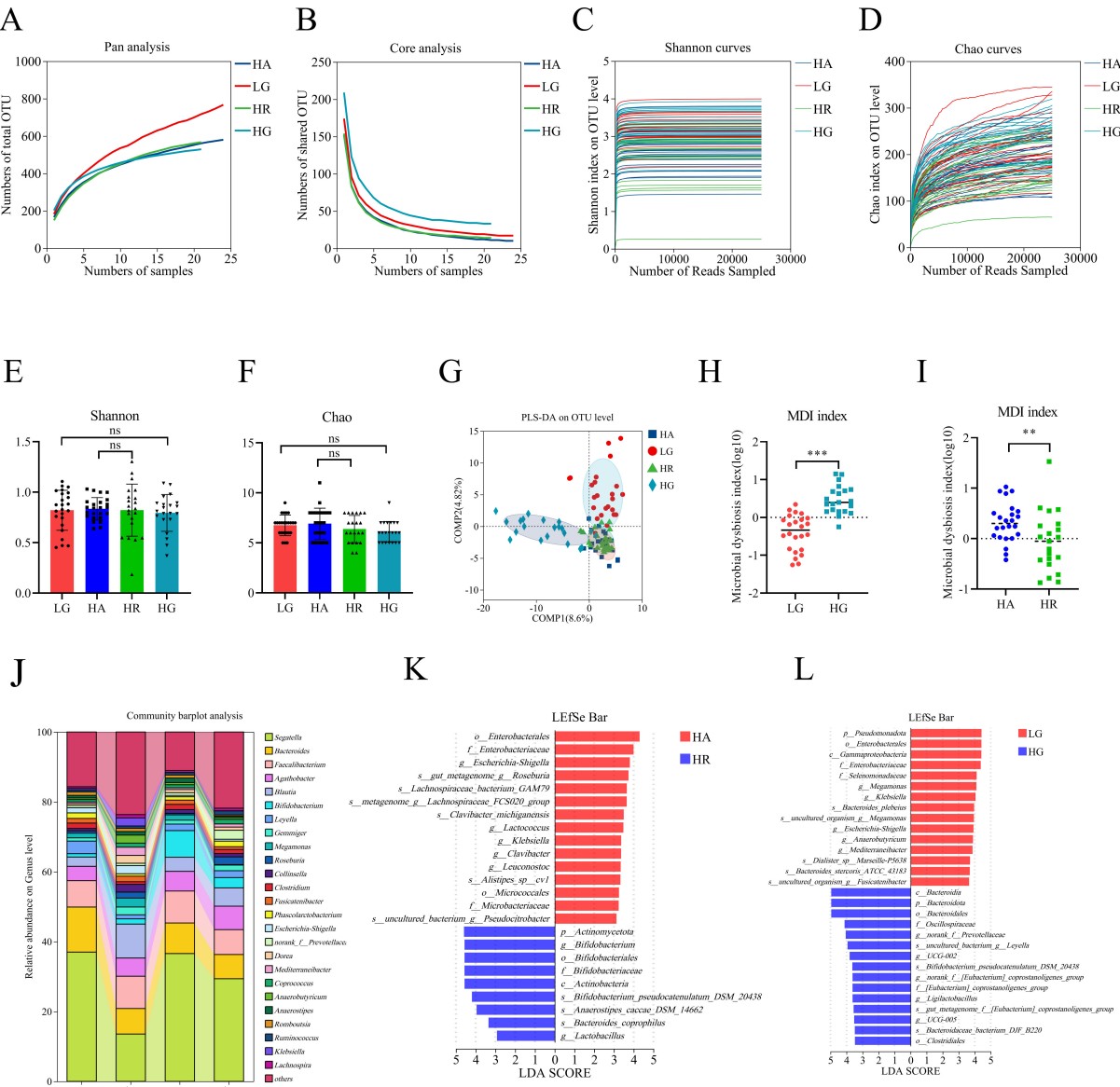

**FIG 1** Hypoxia alters the human gut microbiota. (A, B) Pan/Core analysis. The Pan species curve (A) and Core species curve (B) illustrate the changes in total and core species as the sample size increases. The flatness of the pan/core species curve indicates the adequacy of the sequencing sample size. (C, D) Rarefaction curves based on the Shannon index (C) and the Chao1 index (D) offer valuable insights into the sufficiency of the sequencing data. A leveling off of the curve suggests that the sequencing data are sufficient. (E, F) Comparison of gut microbiota alpha diversity among all groups using the Shannon index (diversity) (E) and Chao1 index (richness) (F). (G) PLS-DA of all groups. Each point represents one sample, and points with the same color and shape belong to the same group. (H, I) Comparison of the MDI between the LG and HG groups (H), as well as between the HA and HR groups (I). The MDI serves as a metric to assess the degree of microbial imbalance, with higher values indicating a greater extent of bacterial disturbance. (J) The species composition at the genus level is illustrated in the bar chart, which displays the top 25 genera based on their relative abundance. (K, L) Results of LEfSe analysis showing significantly differentiated bacterial taxa between different groups. The color red indicates a significant increase in the HA (K) and LG (L) groups, while blue indicates a significant increase in the HR (K) and HG (L) groups. Statistical analysis was performed using one-way ANOVA (E), Kruskal-Wallis test (F), or Student's $t$-test (H, I). **$P < 0.01$, ***$P < 0.001$; "ns" indicates no significance. Data are means ± SD.

## The optimal prebiotics for *B. longum* and *L. plantarum* and their combined inhibition of the growth of *E. coli* and *K. pneumoniae*

This study was conducted to determine the most effective prebiotics for *Lactiplantibacillus* and *Bifidobacterium*. Galactooligosaccharides were found to be the most effective prebiotics for *L. plantarum*, followed by soybean oligosaccharide, inulin, and

isomaltooligosaccharide. However, *K. pneumoniae* and *E. coli* also proliferated well with galactooligosaccharides, soybean oligosaccharide, and inulin. Isomaltooligosaccharide had the least stimulating effect on the proliferation of *K. pneumoniae* compared to the other prebiotics, and it had a significantly lower effect on the growth of *E. coli* compared to galactooligosaccharides, soybean oligosaccharide, and inulin. Thus, isomaltooligosaccharide was selected as the optimal prebiotics for *L. plantarum* (Fig. S5A through D). For *B. longum*, fructooligosaccharide was found to be the most effective prebiotic, with a significantly lower effect on the growth of *K. pneumoniae* and *E. coli* compared to soy oligosaccharide and galactooligosaccharides. Therefore, fructooligosaccharide was selected as the optimal prebiotic for *B. longum* (Fig. S5A through D). Subsequently, the inhibitory effects of the synbiotics on the growth of *E. coli* and *K. pneumoniae* were tested *in vitro*. The results demonstrated that *L. plantarum* and *B. longum* significantly reduced the viable counts of *K. pneumoniae* and *E. coli* (Fig. S5E).

## *E. coli* and *K. pneumoniae* contribute to HAPE under hypoxic conditions

To investigate the effect of bacterial transplantation on HAPE, the rats were divided into five groups: normoxia control (NC) group, hypoxia control (HC) group, normoxia + *E. coli* and *K. pneumoniae* (NKE) group, hypoxia + *E. coli* and *K. pneumoniae* (HKE) group, and hypoxia + *E. coli* and *K. pneumoniae* + synbiotics (HKE+S) group (Fig. S2). To verify the colonization of the target strains in the rat gut, we initially conducted a microbiome analysis of stool samples from the rats. The results indicated that the relative abundance of *K. pneumoniae* and *E. coli* was significantly higher in the NKE and HKE groups compared to the NC and HC groups (Fig. 2A and B). However, treatment with synbiotics significantly reduced the relative abundance of *K. pneumoniae* and *E. coli*, while the levels of *Bifidobacterium* and *Lactiplantibacillus* significantly increased (Fig. 2C). These results confirm that the target strains did colonize the rat gut.

Subsequently, we evaluated the degree of HAPE in rats from different groups. The results showed that the rats in the HC, HKE, and HKE+S groups had significantly reduced body weight, food intake, and water intake compared to the NC and NKE groups (Fig. 2D through F). These symptoms indicate that the rats were struggling due to hypoxia and showed signs of anorexia. Interestingly, lung water content and lung tissue damage scores were not significantly different between the HC and NC groups, as well as between the NC and NKE groups. In addition, rats from the HKE group showed a significant increase in lung water content compared to the NC, HC, and NKE groups (Fig. 2G and I). Histological analysis revealed that rats in the HKE group exhibited lung tissue damage, characterized by alveolar and interstitial hemorrhage, thickening of the alveolar interstitium, and mild infiltration of granulocytes, in comparison to lung tissue from the other groups (Fig. 2H). However, after synbiotics treatment, lung water content and lung tissue damage improved significantly (Fig. 2G through I).

## *E. coli* and *K. pneumoniae* induce inflammation under hypoxic conditions

The relationship between bacterial transplantation and inflammation during HAPE was investigated. We found that although there was a trend for increased levels of IL-6 and IL-17 in the HC group compared to the NC group, the differences were not significant. However, MCP-1 levels were significantly increased in the HC group (Fig. 3A through E). Compared to the NC and HC groups, *E. coli* and *K. pneumoniae* transplantation significantly increased the plasma levels of TNF-α, IL-1β, IL-6, and MCP-1. After synbiotics treatment, the levels of these cytokines decreased significantly (Fig. 3A through E). We also found that the level of IL-17 was significantly higher in the HKE group than in the NC group. Although there was an increasing trend in IL-17 levels in the HKE group compared to the HC and HKE+S groups, the difference was not significant (Fig. 3A through E).

Furthermore, we analyzed the transcriptome of the lung tissue, which revealed a significant enrichment of the chemokine signaling pathway and TNF signaling pathway in the HKE group compared to the HC group. Further analysis of differentially expressed genes in these two pathways revealed significantly upregulated expression of the genes

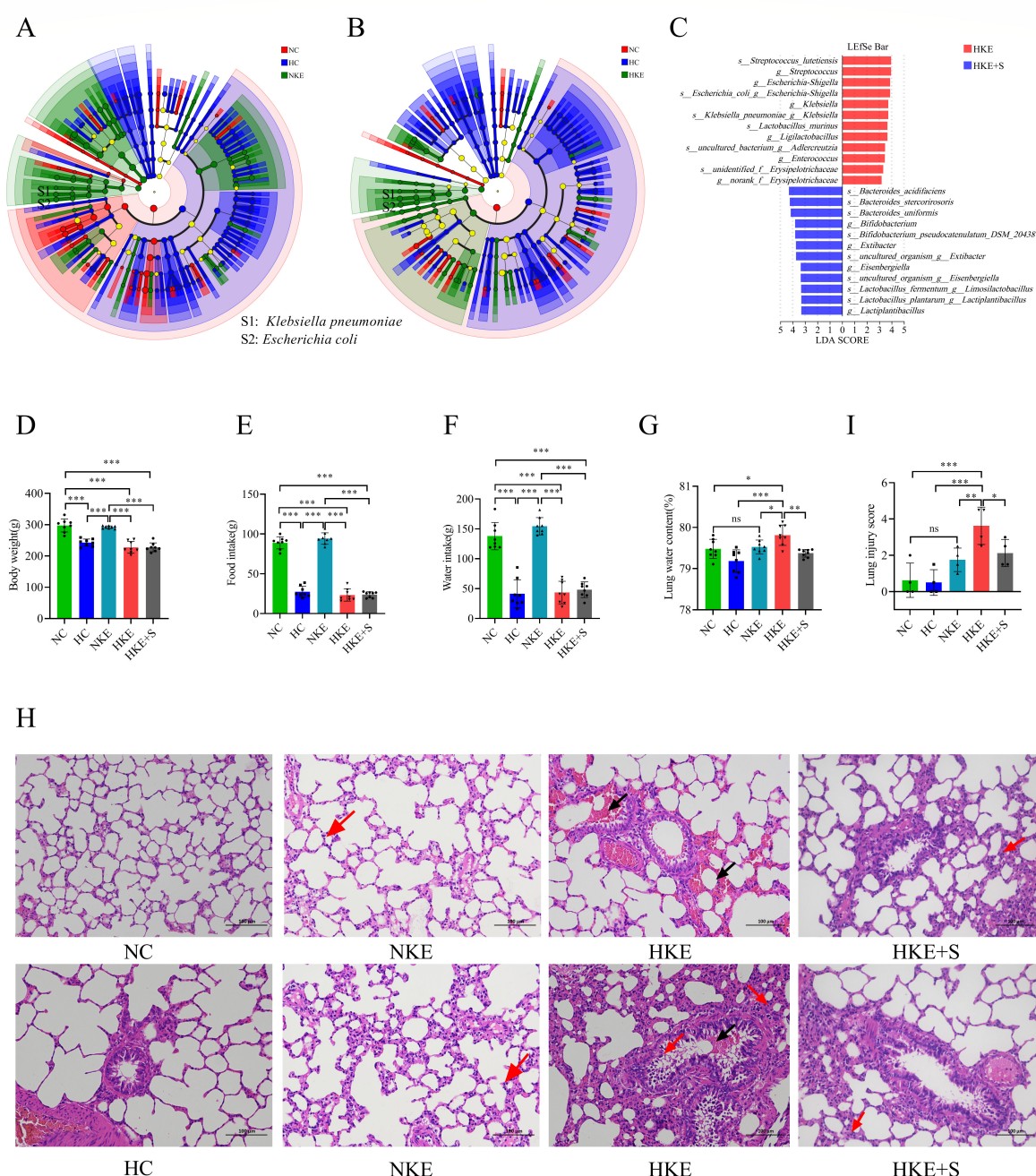

**FIG 2** *E. coli* and *K. pneumoniae* contribute to HAPE under hypoxic conditions. (A, B) Taxonomic cladogram obtained from LEfSe analysis displaying significantly differentiated bacterial taxa. Red indicates a significant increase in the NC group (A, B); blue indicates a significant increase in the HC group (A, B); green indicates a significant increase in the NKE group (A) or the HKE group (B). Nodes labeled S1 and S2 represent species *K. pneumoniae* and *E. coli* (A, B), respectively (*n* = 6). (C) Results of LEfSe analysis showing significantly differential bacterial taxa between the HKE and HKE+S groups (*n* = 6). Red indicates a significant increase in the HKE group, while blue indicates a significant increase in the HKE+S group. (D–G) Comparisons of body weight (D), food intake (E), water intake (F), and lung water content (G) between different treatment groups (*n* = 8). (H) H&E staining of lung tissues. Bar = 100 µm, magnification 200×. Red arrows indicate inflammatory exudation; black arrows indicate hemorrhage. (I) Semi-quantitative histopathological score of lung injury (*n* = 4). Statistical analysis was performed by one-way ANOVA. *P < 0.05, **P < 0.01, ***P < 0.001 (D–G and I). Data are means ± SD.

Ifi47, Vegfd, Irf1, Vcam1, Csf1, Cxcl10, Ccl12, Cx3cr1, Cxcl9, Cxcl16, and Shc4, while only one gene, Plcb2, showed significant downregulation in the HKE group (Fig. 3F). Although there were no significant differences in the 12 genes between the HKE and HKE+S groups, we found that the IL-17 pathway was significantly enriched with significantly

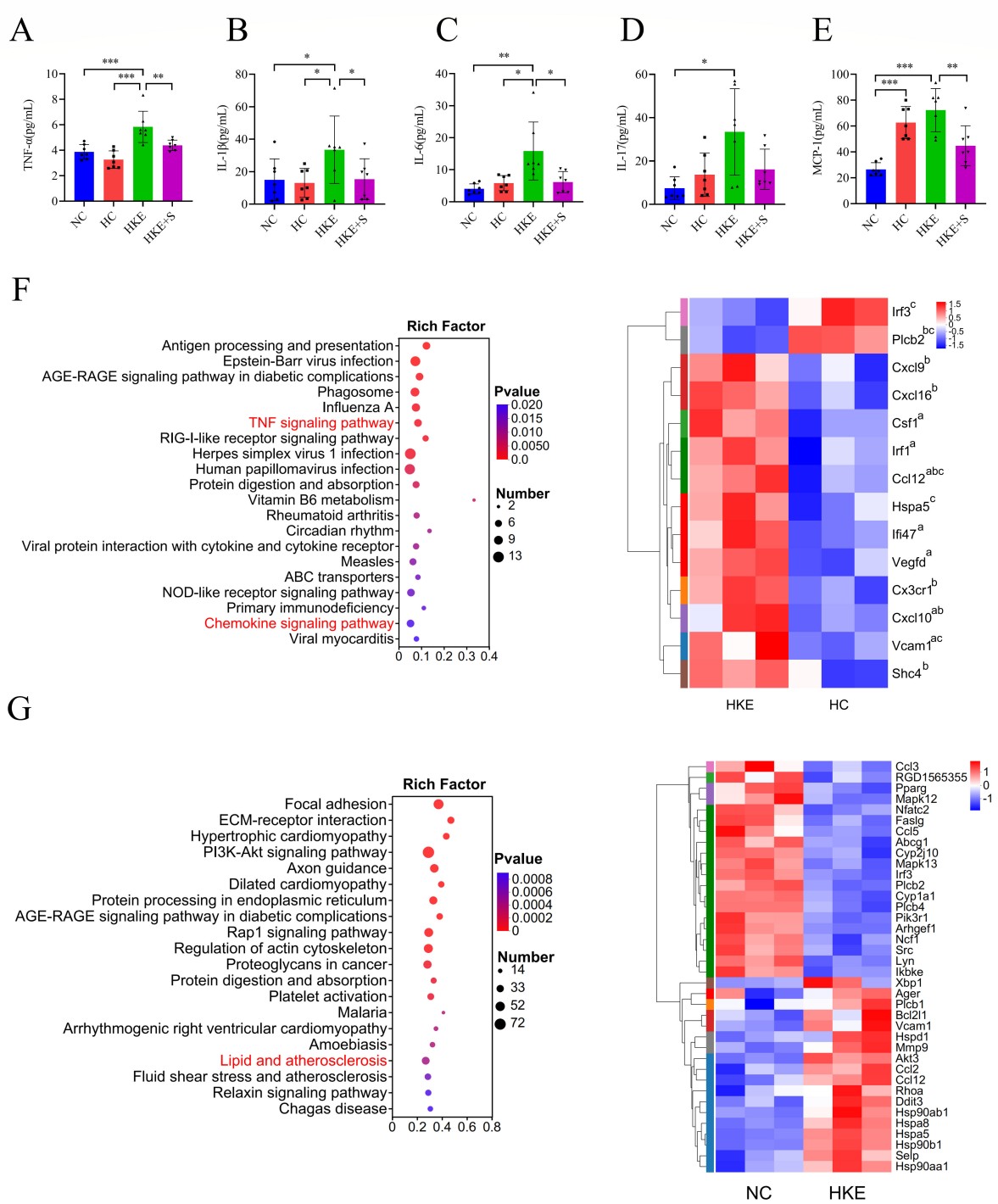

**FIG 3** *E. coli* and *K. pneumoniae* induced inflammation during HAPE under hypoxic conditions. (A–E) Comparisons of TNF-α (A), IL-1β (B), IL-6 (C), IL-17 (D), and MCP-1 (E) levels in rat plasma between different groups (*n* = 7). (F) KEGG pathway enrichment analyses using the differentially expressed genes between the HKE and HC groups, and heat maps showing the differentially expressed genes in the chemokine signaling pathway, TNF signaling pathway, and lipid and atherosclerosis pathway. Genes labeled a, b, and c represent differential expression in the TNF signaling pathway, chemokine signaling pathway, and lipid and atherosclerosis pathway, respectively. (G) KEGG pathway enrichment analyses using the differentially expressed genes between the HKE and NC groups, and heat maps showing the differentially expressed genes in the lipid and atherosclerosis pathway. Statistical analysis was performed by one-way ANOVA (A, B, and E) or Kruskal-Wallis test (C, D). *$P < 0.05$, **$P < 0.01$, ***$P < 0.001$. Data are means ± SD.

decreased expression of the genes Ccl11 and Cebpb in the HKE+S group (Fig. S6A). Interestingly, between the NC and HKE groups, we found 26, 18, and 16 differentially expressed genes related to the chemokine signaling pathway, TNF signaling pathway, and IL-17 signaling pathway, respectively (Fig. S6B and C). In addition, the lipid and atherosclerosis pathway was significantly enriched between the NC and HKE groups, containing 38 differentially expressed genes (Fig. 3G). Similarly, between the HC and HKE groups, five differentially expressed genes were related to the lipid and atherosclerosis pathway (Fig. 3F). We know that the lipid and atherosclerosis pathway is related to lipid metabolism.

## *E. coli* and *K. pneumoniae* disrupt lipid metabolism under hypoxic conditions

Based on transcriptome analysis, the lipid and atherosclerosis pathway was significantly enriched after *E. coli* and *K. pneumoniae* transplantation. Lipidomics analysis was performed to identify the biomarkers involved in HAPE. The PLS-DA score plots showed a clear separation between the HC and HKE groups, HC and HKE+S groups, and HKE and HKE+S groups (Fig. 4A and C; Fig. S7A), indicating significant changes in lipid metabolism after different treatments. Further analysis revealed that the HKE group had 203 significantly altered lipid metabolites compared to the HC group. All 12 altered components of the LPCs were significantly upregulated (Fig. 4B), and 19 out of 25 significantly altered components of the PEs were also upregulated (Fig. S7C). A combined correlation analysis revealed significant positive correlations between most of the LPCs. Specifically, seven LPCs showed significant positive correlations with *Klebsiella*, and nine LPCs exhibited significant positive correlations with *Escherichia-Shigella* (Fig. 4E). Compared to the HC group, six components of LPCs were significantly upregulated in the HKE+S group, representing a 50% decrease in the number of significantly upregulated components of LPCs compared to the number of significantly upregulated LPCs between the HC and HKE groups (Fig. 4D). In addition, one component of LPCs and six components of PEs were found to be significantly lower in the HKE+S group, and there were no significant increases in the components of LPCs and PEs compared to the HKE group (Fig. S7B). These results suggest that LPCs and PEs may contribute to HAPE in rats, while synbiotics may play a beneficial role in ameliorating the disruption of lipid metabolism.

The KEGG pathway enrichment analysis of the significantly different lipids between the HC and HKE groups showed that natural killer cell-mediated cytotoxicity, B cell receptor signaling pathway, T cell receptor signaling pathway, Th17 cell differentiation, and Th1 and Th2 cell differentiation were significantly enriched. Additionally, the chemokine signaling pathway and NF-kappa B signaling pathway were significantly enriched, indicating that dyslipidemia induces inflammation during HAPE (Fig. 4F). The level of diacylglycerol (DG; 18:0/16:0) associated with the above pathways was significantly increased after transplantation of *E. coli* and *K. pneumoniae* (Fig. 4B). This suggests that DG (18:0/16:0) is involved in immune regulation during HAPE.

## LPCs cause damage and permeability to HPMEC and HPAEpiC monolayers

We conducted a study to investigate the detrimental effects of LPCs and PEs on HPMEC and HPAEpiC monolayers. Our results showed that LPCs significantly reduced the viability of both HPAEpiC and HPMEC monolayers in a concentration-dependent manner under hypoxic conditions (4% $O_2$). The damaging effect of LPCs was observed to be stronger on HPMEC monolayers than on HPAEpiC monolayers (Fig. 5A). Similarly, PEs significantly reduced the viability of HPMEC monolayers. However, no significant deleterious effect of PEs was observed on HPAEpiC monolayers (Fig. 5B). We then tested the effect of LPCs on the permeability of HPMEC and HPAEpiC monolayers. The results showed that LPCs increased the permeability of both types of cell monolayers in a concentration-dependent manner at 4% oxygen concentration (Fig. 5C).

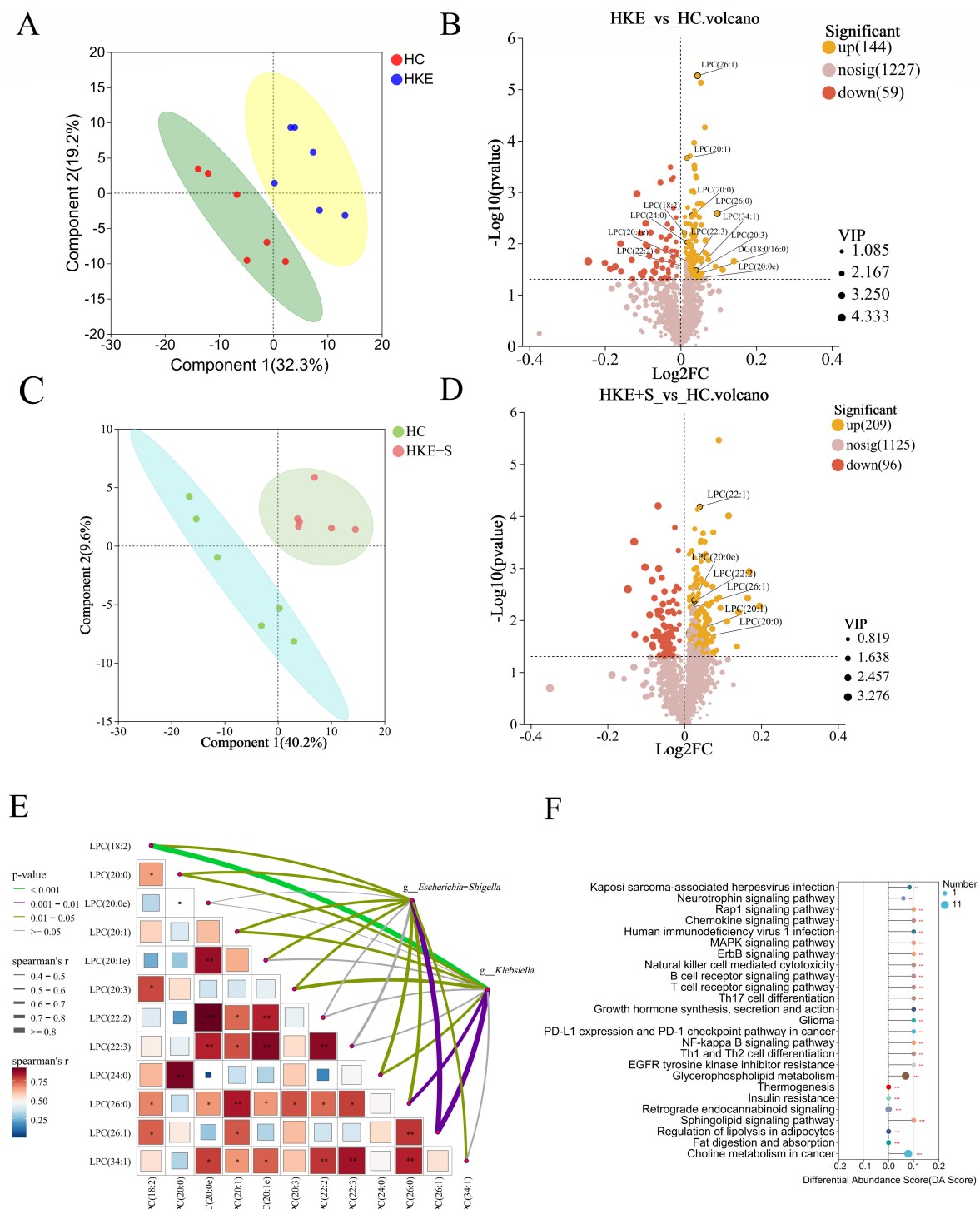

**FIG 4** *E. coli* and *K. pneumoniae* disrupt lipid metabolism under hypoxic conditions. (A, C) PLS-DA between the HC and HKE groups (A), as well as between the HC and HKE+S groups (C). Each point represents one sample, and points of the same color belong to the same group. (B, D) Volcano plot of differential metabolites between HKE and HC (B), as well as between HKE+S and HC (D). (E) The heat map displays the pairwise Spearman's correlation matrix of the 12 differential LPCs between the HKE and HC groups. The color gradient represents Spearman's correlation coefficients, while *, **, and *** indicate significant correlations at $P <$ 0.05, $P < 0.01$, and $P < 0.001$, respectively. The relationships of the differential LPCs with *E. coli* and *K. pneumoniae* were assessed using Spearman correlation analysis. The width of the borders reflects the Spearman's r statistic, and the color of the borders indicates statistical significance. (F) KEGG pathway-based differential abundance (DA) scoring between the HC and HKE groups. The horizontal axis represents the DA score, while the vertical axis indicates the names of the KEGG pathways. The size of each dot reflects the number of differential metabolites associated with that pathway; larger dots signify a greater quantity of

Fig 4 (Continued)

differential metabolites. When the dots are placed to the right of the central axis, the longer the line segment, the more the overall expression of the pathway tends to be upregulated. Conversely, when the dots are placed to the left of the central axis, the longer the line segment, the more the overall expression of the pathway tends to be downregulated.

## LPCs disrupt membranes of HPMEC and HPAEpiC

To understand how LPCs reduce the viability of HPMECs and HPAEpiCs, we examined the rescue effects of seven commonly used inhibitors. These inhibitors include Z-VAD-FMK (Z-VAD), nifedipine (Nif), glutathione (GSH), chloroquine (CQ), necrostatin-1 (Nec-1), liproxstatin-1 (Lip-1), and ferroxstatin-1 (Fer-1). Our results showed that none of the seven inhibitors could reverse the damaging effects of LPCs on HPAEpiCs cells (Fig. 5D). Although chloroquine could significantly reduce cell viability loss in HPMECs cells, the increase in viability was minimal, at only 5%. Furthermore, none of the other inhibitors could rescue the cell death induced by LPCs (Fig. 5E).

To determine whether LPCs could cause direct damage to cell membranes, we measured lactate dehydrogenase levels and observed cell membranes by scanning electron microscopy. The results showed that LPCs caused a concentration-dependent increase in lactate dehydrogenase levels in both HPAEpiC and HPMEC monolayers, indicating membrane damage (Fig. 5F). In addition, scanning electron microscopy revealed an increased number of abnormal cells and the presence of pores and disruptions on cell membranes after treatment with LPCs (Fig. 6A). These findings suggest that LPCs have the potential to damage cell membranes, which could ultimately lead to cell death.

## Molecular dynamics simulation of the interaction process between LPCs and cell membranes

To observe how LPCs interact with cell membranes, two concentrations of LPCs were randomly placed over the cell membrane at a distance of 1 to 5 nm, and molecular dynamics simulations were performed for 100 ns. The results showed that even at low concentrations of LPCs (6.25 µg/mL), small pores appeared on the cell membrane (Fig. S8). At high concentrations of LPCs (100 µg/mL), LPC molecules inserted into the cell membrane, aggregated, and disrupted the integrity of the lipid bilayer, forming pores. Over time, these pores increased in size, compromising the barrier function of the membrane, which could lead to uncontrolled exchange of substances inside and outside the cell (Fig. 6B). The formation of these pores not only provides direct evidence that LPCs disrupt membrane integrity, but also indicates changes in membrane fluidity and lipid rearrangements that may contribute to cell death.

## LPCs promote HAPE in mice under hypoxic conditions

To investigate the role of LPCs in the development of HAPE, we administered a dose of 10 mg/kg of LPCs to mice via the tail vein and exposed them to a simulated altitude of 6,500 m. The results indicated that mice in the hypoxia + LPC (H+LPC) group exhibited increased lung water content and elevated lung tissue damage scores compared to the hypoxia control (HC) group (Fig. 7A and C). This damage was characterized by interstitial pulmonary edema, thickening of the alveolar interstitium, and granulocyte infiltration when compared to the lung tissue of the normoxia control (NC) and HC groups (Fig. 7B). However, there was no significant difference in lung water content or lung tissue damage scores between the NC and normoxia + LPC (N+LPC) groups (Fig. 7). These findings suggest that LPCs play a crucial role in the development of HAPE specifically under high-altitude conditions.

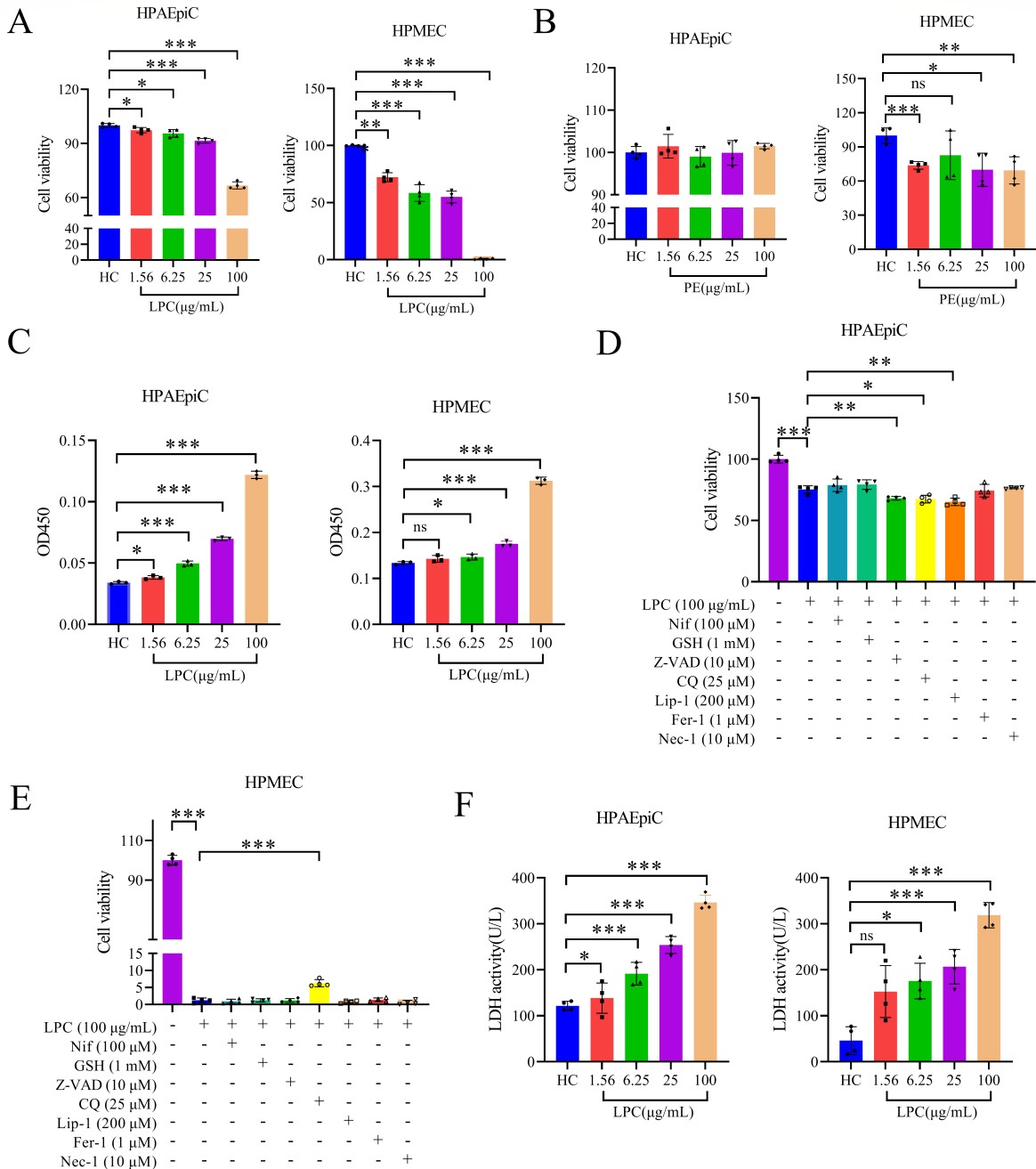

**FIG 5** LPCs cause damage and permeability to HPMEC and HPAEpiC monolayer. (A) HPMEC and HPAEpiC monolayers were treated with different concentrations of LPCs, and cell viability was assessed (*n* = 4). (B) HPMEC and HPAEpiC monolayers were treated with different concentrations of PEs, and cell viability was determined (*n* = 4). (C) HPMEC and HPAEpiC monolayers were treated with different concentrations of LPCs, and permeability was determined (*n* = 3). (D) HPAEpiCs were treated with 100 µg/mL LPCs, either with or without various inhibitors, and cell viability was assessed (*n* = 4). (E) HPMECs were treated with 100 µg/mL LPCs, with or without different inhibitors, and cell viability was evaluated (*n* = 4). (F) HPMEC and HPAEpiC monolayers were treated with different concentrations of LPCs, and LDH release assay was performed (*n* = 4). Statistical analysis was performed by Student's *t*-test (A, C through F). Kruskal-Wallis test was applied to compare the HC group with the 1.56 µg/mL LPC group (HPAEpiCs), while the Student's *t*-test was utilized for the remaining comparisons (B). *$P <$ 0.05, **$P <$ 0.01, ***$P <$ 0.001; "ns" indicates no significance. Data are means ± SD.

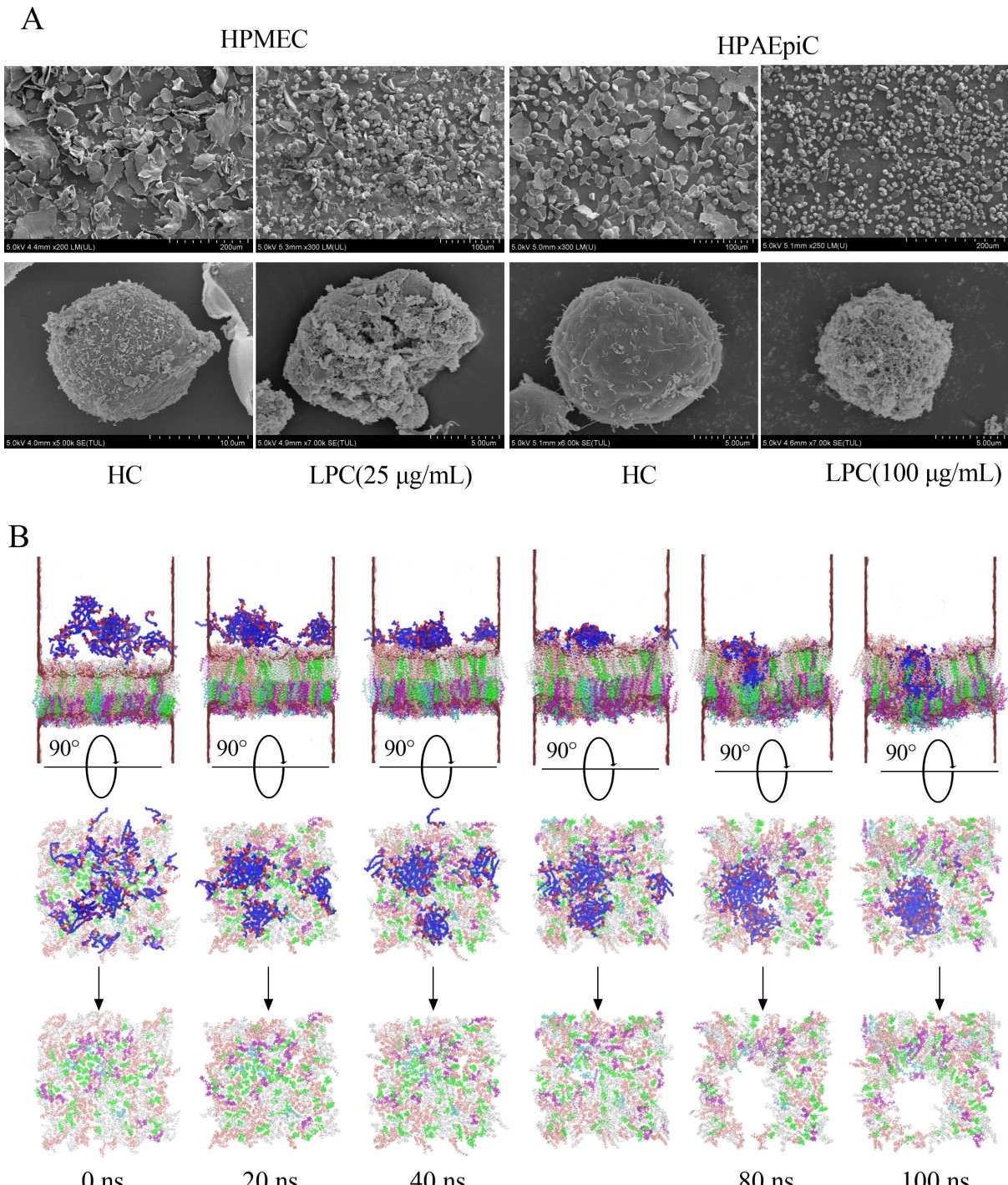

**FIG 6** LPCs disrupt the cell membranes. (A) SEM images of HPMECs and HPAEpiCs after different treatments. (B) Molecular dynamics simulation of the interaction process between LPCs and cell membranes. The cell membrane is composed of a variety of lipids represented by different colors: green for CHL1, white for PSM, pink for POPC, cyan for POPS, and purple for POPE. LPC molecules are shown in blue and are randomly distributed in the upper part of the membrane. Water molecules are shown in a cotton-like transparent shape that surrounds the outside of the entire system.

## DISCUSSION

The specific pathogens involved in HAPE and the underlying mechanisms are still unknown. As HAPE can cause fatal injuries in volunteers, we primarily investigated the composition of the gut microbiota during AMS to predict the different species associated

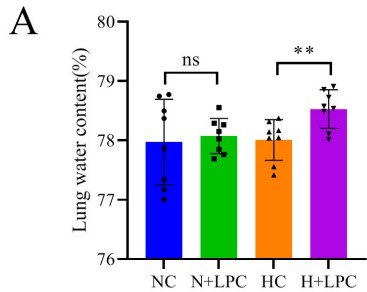

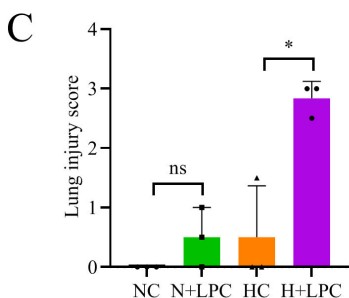

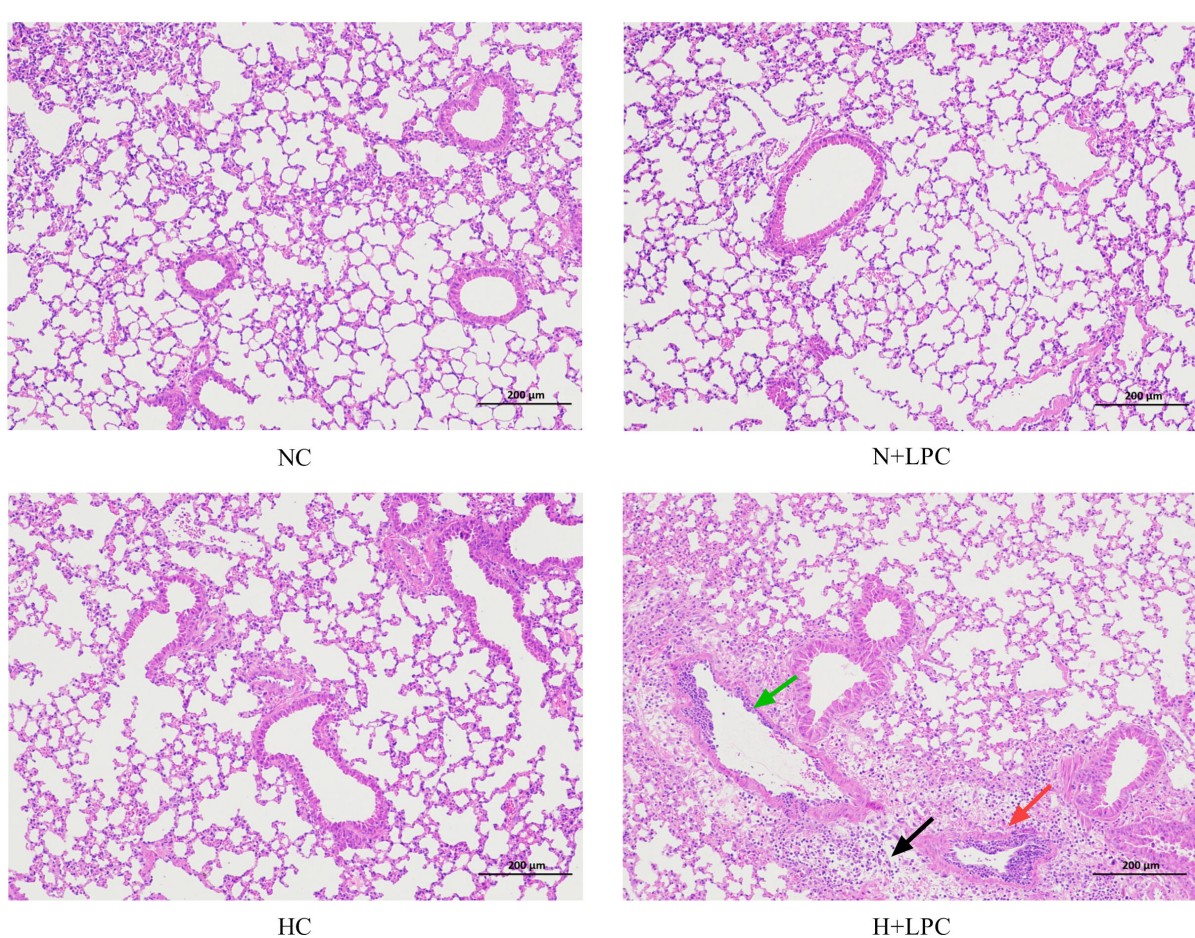

**FIG 7** LPCs promote HAPE in mice under hypoxic conditions. (A) Lung water content across different treatment groups was analyzed ($n = 8$). (B) H&E staining of lung tissues. Bar = 200 µm, magnification 100×. Red arrows indicate inflammatory exudation; green arrows denote the accumulation of white blood cells against the walls of blood vessels; and black arrows indicate interstitial pulmonary edema. (C) Semi-quantitative histopathological score of lung injury ($n = 3$). Statistical analysis was conducted using either the Student's $t$-test (A) or the Mann-Whitney U test (C). *$P < 0.05$, **$P < 0.01$; "ns" indicates no significance. Data are presented as means ± SD.

with HAPE. The results showed a significant decrease in the relative abundance of *Klebsiella* and *Escherichia-Shigella* and a significant increase in the relative abundance of *Bifidobacterium* and Lactobacillus in volunteers when AMS disappeared (Fig. 1K). We postulate that *Klebsiella* and *Escherichia-Shigella* may be pathogenic, contributing to HAPE, while *Bifidobacterium* and *Lactobacillus* may be beneficial in preventing HAPE.

In addition to *Escherichia-Shigella* and *Klebsiella*, our gut microbiota profiling identified several other microbes, including *Lactococcus*, *Clavibacter*, and *Leuconostoc*. These microbes exhibited significant increases in volunteers who experienced severe AMS. However, their average relative abundances were low, with *Lactococcus* at 0.0036%, *Clavibacter* at 0.0026%, and *Leuconostoc* at 0.0026%. Therefore, we hypothesize that these microbes only play a negligible role in HAPE. Further research showed that transplantation of *K. pneumoniae* and *E. coli* promoted HAPE in rats exposed to high altitude, while a synbiotic composed of *Bifidobacterium*, *Lactobacillus plantarum*, isomaltose oligosaccharide, and fructooligosaccharide significantly reduced the severity of this condition (Fig. 2G through I). Interestingly, rats transplanted with *E. coli* and *K. pneumoniae* did not exhibit HAPE when maintained at low altitudes. Therefore, we speculate that gut opportunistic pathogens must interact with various factors induced by high-altitude exposure, such as inflammation, pulmonary hypertension, and intestinal barrier damage, to contribute to the development of HAPE. This study found that there was no significant difference in lung water content between the hypoxia control and normoxic control groups. This parallels findings in a high-altitude cerebral edema model (50). Additionally, there is no single standard for the animal model of high-altitude pulmonary edema (51), and the modeling conditions reported in the literature vary (30, 52, 53). This study also discovered that it was difficult to induce high-altitude pulmonary edema in rats without administering *E. coli* and *K. pneumoniae*. Based on these findings, intestinal *K. pneumoniae* and *E. coli* were discovered as the opportunistic pathogens of HAPE, which will be vital for understanding the development of this disease.

This study showed that transplantation of *K. pneumoniae* and *E. coli* increased cytokine levels in rat plasma (Fig. 3A through E). The chemokine signaling pathway and TNF signaling pathway were significantly enriched (Fig. 3F), and seven differentially expressed genes in the TNF signaling pathway were significantly upregulated after *K. pneumoniae* and *E. coli* transplantation. To exclude the possibility of *E. coli* and *K. pneumoniae* infection, 100 µL of rat blood and lung tissue homogenates from the NKE and HKE groups were cultured on basal salt medium containing 1% glucose, and no culturable colonies were observed (Fig. S9). These results confirm that the administration of *E. coli* and *K. pneumoniae* does not trigger a systemic response or lead to the spread of these bacteria from the site of inoculation to the lungs. Although many studies have suggested that inflammation contributes to the pathogenesis of HAPE (54, 55), the relationship between inflammation during HAPE and gut microbiota has not been reported. In this study, inflammation during HAPE was found to be induced by the gut microbiota, further demonstrating the important role of the gut microbiota in HAPE. In addition, we observed that the lipid and atherosclerosis pathway was enriched with 38 differentially expressed genes between the NC and HKE groups (Fig. 3G) and five differentially expressed genes between the HC and HKE groups related to this pathway, indicating disruption of lipid metabolism.

Lipidomics analysis revealed that the transplantation of *E. coli* and *K. pneumoniae* causes disruptions in lipid metabolism in rat plasma, leading to the significant upregulation of a large number of LPCs (Fig. 4B). Correlation analysis showed that *Escherichia-Shigella* and *Klebsiella* were associated with increased levels of LPCs (Fig. 4E). To explore the potential mechanism by which gut pathogens are involved in the elevation of LPCs, we searched the KEGG database and found that *E. coli* and *K. pneumoniae* can express phospholipase A2, an enzyme that metabolizes intestinal phospholipids into LPCs, which may be related to the elevated plasma levels of LPCs. Interestingly, research indicates that the absence of PLA2g1b in the intestinal lumen significantly reduces the absorption of LPCs, which suggests that PLA2g1b plays a role in increasing plasma LPC levels (56). In contrast, *B. longum* and *L. plantarum* do not express this enzyme. In this study, LPCs were found to cause damage and permeability to HPMECs and HPAEpiCs under hypoxic conditions (Fig. 5A and C), indicating that LPCs may cause pulmonary microvascular leakage and further lead to HAPE. Although LPC has been implicated as a critical factor in promoting atherosclerosis and cardiovascular disorders, its role in

HAPE has not been described (57). In this study, the mouse experiment provided further evidence that LPC promotes HAPE (Fig. 7). Notably, a study involving a human cohort found that all five altered components of the LPCs were significantly upregulated in subjects susceptible to AMS compared to those resistant to AMS (58). Interestingly, the chemokine and NF-kappa B signaling pathways were enriched (Fig. 4F), suggesting that dyslipidemia is involved in the regulation of inflammation. The level of DG (18:0/16:0) associated with the above pathways significantly increased after the transplantation of *E. coli* and *K. pneumoniae*. The DG (18:0/16:0) induces inflammation through the NF-kappa B signaling pathways, according to the KEGG pathway analysis. This suggests that gut pathogen-induced DG (18:0/16:0) promotes inflammation during HAPE.

Previous studies have suggested that LPCs induce cell death through apoptosis and autophagy (59, 60), but by blocking these cell death pathways with appropriate inhibitors, we found that these pathways do not play a major role in LPC-induced cell death. However, our results showed that LPCs induce cell necrosis mainly by inserting and aggregating into the cell membrane and disrupting membrane integrity (Fig. 6A). This different finding suggests that a membrane stabilizer (61) may be resistant to the destructive effect of LPC on cell membranes and may be beneficial in HAPE and other diseases associated with LPC.

Overall, our results show that intestinal *K. pneumoniae* and *E. coli* are the opportunistic pathogens of HAPE, that gut microbiota-induced elevation of LPCs and inflammation promote HAPE, and that a synbiotic has a good preventive effect against HAPE. We hypothesize that gut microbiota and LPCs may be areas of interest for future research in acute altitude illness. The metabolic pathways and changes in host physiology that they influence may inform preventive measures that could benefit populations living at high altitude or traveling from low to high altitude. However, further studies or clinical trials are needed to confirm these findings and to improve prevention and treatment recommendations for HAPE.

## ACKNOWLEDGMENTS

We thank the low-altitude volunteers of the SPU Scientific Expedition of TCM Resources and the high-altitude volunteers for their selfless participation in this study; Mr. Dorje Shala, Mr. Xiaoming Dong, and Mr. Jian Huang for their thoughtful arrangements during the volunteers' stay at high altitudes; Mr. Jimin Bao, Mr. Li Ding, and Ms. Jiaojiao Li from Jinqiu Hospital of Liaoning Province for providing medical care to the volunteers; Professor Chunfu Wu and Professor Lihui Wang from Shenyang Pharmaceutical University, and Director Zhihui Hao from the Experimental Animal Center, for their experimental suggestions.

This work was supported by the Professional Technology Innovation Centre for Development and Application of Endangered and Rare Medicinal Material Resources of Liaoning Province, the Engineering Laboratory for Development and Application Technology of Authentic and Endangered Rare Medicinal Materials of Liaoning Province, and the Benxi Drug Research Institute of Shenyang Pharmaceutical University. This work was supported by the People's Hospital of Maqin County, Guoluo Prefecture, Qinghai Province. This work was supported by the Foundation of Shenyang Pharmaceutical University (52104003 and 51110656).

X.S., conceptualization, data curation, formal analysis, investigation, methodology, software, visualization, writing – original draft, writing – review and editing; G.H., data curation, resources; Y.L., validation; W.L., validation, data curation; Y.W., validation; H.Y., validation; G.L., validation; L.Z., methodology; A.W., project administration, validation; J.J., conceptualization, funding acquisition, project administration, supervision, visualization, writing – review and editing.

## AUTHOR AFFILIATIONS

[1]Department of Traditional Chinese Materia Medica, Shenyang Pharmaceutical University, Shenyang, Liaoning, China
[2]Department of Pharmacology, Shenyang Pharmaceutical University, Shenyang, Liaoning, China

## AUTHOR ORCIDs

Anhua Wang http://orcid.org/0000-0002-8615-3424
Jingming Jia http://orcid.org/0009-0002-0519-0854

## AUTHOR CONTRIBUTIONS

Xianduo Sun, Conceptualization, Data curation, Formal analysis, Investigation, Methodology, Software, Visualization, Writing – original draft, Writing – review and editing | Gaosheng Hu, Data curation, resources | Yuting Li, Validation | Wenjing Li, Data curation, Validation | Yong Wang, Validation | Hui Yan, Validation | Guoqing Long, Validation | Long Zhao, Methodology | Anhua Wang, Project administration, Validation | Jingming Jia, Conceptualization, Funding acquisition, Project administration, Supervision, Visualization, Writing – review and editing

## DATA AVAILABILITY

The 16S rRNA gene sequence data and RNA-seq data have been submitted to the NCBI Sequence Read Archive (SRA) database under the accession numbers PRJNA1238177 and PRJNA1119188, respectively. The untargeted lipidomics data are available in the Zenodo database at https://doi.org/10.5281/zenodo.15062151.

## ETHICS APPROVAL

All research procedures involving human subjects were approved by the Ethics Committee of Maqin County People's Hospital, Golog Prefecture, Qinghai Province (Study No. MQXrmyyll-2019-001). Written informed consent was obtained from all volunteers. The animal study was conducted following the Guideline of Animal Experimentation of Shenyang Pharmaceutical University and was approved by the Animal Ethics Committee of the institution (approval number: SYPU-IACUC-2019-1227-205).

## ADDITIONAL FILES

The following material is available online.

### Supplemental Material

**Supplemental material (Spectrum03057-24-s0001.pdf).** Fig. S1 to S9; Tables S1 and S2.

### Open Peer Review

**PEER REVIEW HISTORY (review-history.pdf).** An accounting of the reviewer comments and feedback.

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
