## [Reviewer comments · Microbiology Spectrum]

Microbiology Spectrum

Gut opportunistic pathogens contribute to high altitude pulmonary edema by elevating lysophosphatidylcholines and inducing inflammation

Xianduo Sun, Gaosheng Hu, Yuting Li, Wenjing Li, Yong Wang, Hui Yan, Guoqing Long, Long Zhao, Anhua Wang, and jingming Jia

Corresponding Author(s): jingming Jia, Shenyang Pharmaceutical University

Review Timeline:

Submission Date:	November 25, 2024
Editorial Decision:	January 25, 2025
Revision Received:	March 23, 2025
Accepted:	April 15, 2025

Editor: Zhe LYU

Reviewer(s): Disclosure of reviewer identity is with reference to reviewer comments included in decision letter(s). The following individuals involved in review of your submission have agreed to reveal their identity: Jinshan Suo (Reviewer #2)

Transaction Report:

DOI: <https://doi.org/10.1128/spectrum.03057-24>

Re: Spectrum03057-24 (**Gut opportunistic pathogens contribute to high altitude pulmonary edema by elevating lysophosphatidylcholines and inducing inflammation**)

Dear Prof. jingming Jia:

Thank you for the privilege of reviewing your work. Below you will find my comments, instructions from the Spectrum editorial office, and the reviewer comments.

I agree with the reviewers that your work has merit. However, both reviewers also raised substantial concerns with regard to sample size, methodology, and confounders. Please take your time and address these concerns thoroughly.

Revision Guidelines

Sincerely,
Zhe LYU
Editor
Microbiology Spectrum

Reviewer #1 (Comments for the Author):

The article by Sun and colleagues investigates the role of the human gut microbiota in high-altitude pulmonary edema (HAPE) and acute mountain sickness (AMS). The authors report that the gut microbiota is altered in acute mountain sickness and that the relative abundance of *Klebsiella* sp., *E. coli* and *Shigella* sp. was lower in subjects on recovery along with an increase in the

abundance of Bifidobacterium sp. Based on this finding, they have isolated strains of K. pneumoniae, E. coli, Bifidobacterium sp. etc. and conducted animal experiments to propose that such alterations in the gut microbiota (increased E. coli and K. pneumoniae) can determine host responses to simulated high altitude. They also show that treatment of E. coli and K. pneumoniae colonized animals with synbiotics made of Bifidobacterium sp. and prebiotics rescues from the effects of HAPE.

I have the following major concerns regarding the experiments performed:

Microbiota analysis:

1. For 16S rRNA-based gut microbiota profiling, the authors have not indicated the average number of reads generated per sample. This information is critical and will inform whether they have achieved sequencing depth to draw the conclusions. They could also plot a rarefaction plot to indicating sufficient sequencing depth for samples studied.
2. Apart from E. coli and Klebsiella sp., the gut microbiota profiling showed altered levels of several other microbes. Their potential role has not been discussed in the manuscript.
3. I do not agree with the response of the authors to Q8 raised by the reviewer. The method indicated is not an appropriate way of determining sample size for a gut microbiota study.
4. The pipeline (QIIME1) and database used for analysis (Silva128) are quite outdated and more recent versions with better functionality are available. It is currently unclear if this affects the study findings.

Animal experiments:

1. Given that the isolates are human derived, how are the authors sure that the strains colonized the animal gut? Given that colonization is largely influenced by the host and its diet, human-derived strains may not have colonized the animal at all. Furthermore, administration through the oral route may affect delivery to the gut. Thus, gut microbiota data for animal models after inoculation and treatments may provide better clarity.
2. Have the authors considered that the E. coli and K. pneumoniae used in their study could be pathogenic and may trigger a systemic response? Also, K. pneumoniae can easily disseminate from the site of inoculation to distal body sites, triggering responses there.
3. The methods do not mention the actual doses of the microbes delivered. The dosage and the rationale for selection of dosage needs to be indicated in the manuscript.
4. As per the methods, 20 ml of the synbiotic ($\sim 10^9$ CFU/ml) was administered daily for 3 days. Again, what was the rationale for selection of this dosage.
5. The control suggested by Reviewer 1 in Q5 (NC + KE control) is critical and the authors have not implemented that same.
6. In Fig. 2D, the difference is very marginal and there is no need to have a cut graph to highlight the marginal difference.

Cell culture experiments:

1. The query raised by Q4 by Reviewer 1 also remains unanswered. It is unlikely that LPCs can accumulate at tested (of 25 $\mu\text{g}/\text{mL}$ 100 $\mu\text{g}/\text{mL}$) concentration ranges under physiological conditions. The cited manuscript indicates that LPC levels are higher in AMS-sensitive group, but does not indicate concentrations. This issue is still unresolved.

Other minor comments:

1. The manuscript was submitted without line numbers, which made it difficult to review.
2. The introduction mentions shotgun metagenomics, while the analysis carried out was 16S-based microbiota profiling.
3. Figure legends can still be improved to include more details on the experimentation and statistics, which will make them self-sufficient.

Reviewer #2 (Comments for the Author):

This paper reveals a gut microbiota - LPCs/inflammation - HAPE axis. It discovers the role of gut opportunistic pathogens in HAPE, and how they contribute by elevating LPCs and inducing inflammation. Also, it shows the effects of LPCs on cell monolayers and the preventive effect of a synbiotic on HAPE, which are new insights for understanding and treating HAPE. The article overall is a comprehensive study, but it has some potential limitations. Firstly, the sample size of volunteers in each group is relatively small (only 15 in some groups), which may limit the statistical power and generalizability of the results. Secondly, the animal models used may not fully mimic the complex pathophysiology of HAPE in humans. Thirdly, although the study proposes a gut microbiota - LPCs/inflammation - HAPE axis, there may be other factors or pathways not explored that could also be involved in the development of HAPE.

Dear Editors and Reviewers,

Thank you for your consideration and for the valuable comments provided by the reviewers regarding our manuscript entitled “Gut opportunistic pathogens contribute to high altitude pulmonary edema by elevating lysophosphatidylcholines and inducing inflammation” (Spectrum03057-24). We greatly appreciate these insightful remarks, which have significantly aided us in revising and enhancing our paper, as well as providing important guidance for our research. We have carefully reviewed the comments and made corrections that we hope will meet your approval. The revised portions are highlighted in yellow in the Marked-Up Manuscript. Below are the main corrections made in the paper, along with our responses to the reviewers’ comments:

Reviewer #1 (Comments for the Author):

Microbiota analysis:

1. For 16S rRNA-based gut microbiota profiling, the authors have not indicated the average number of reads generated per sample. This information is critical and will inform whether they have achieved sequencing depth to draw the conclusions. They could also plot a rarefaction plot to indicating sufficient sequencing depth for samples studied.

Response:

We sincerely thank the reviewer for highlighting this important information. Following data optimization, a total of 7,410,203 reads were obtained from the fecal samples, with an average of $82,336 \pm 59,224$ reads per sample. This essential information has been incorporated into the revised manuscript and is highlighted in yellow in the Marked-Up Manuscript (line numbers: 435-437). Additionally, the rarefaction curves based on the Chao and Shannon indices (Figure 1C, D) have been plotted in the revised manuscript. The results demonstrate that as the number of sequences increased, the curves gradually leveled off, indicating saturated coverage

and sufficient sequencing depth for the experiment (Reference 1).

Reference:

1. Cui L, Zhao T, Hu H, et al. Association study of gut flora in coronary heart disease through high-throughput sequencing. *BioMed research international*, 2017, 2017(1): 3796359. <https://doi.org/10.1155/2017/3796359>

2. Apart from *E. coli* and *Klebsiella* sp., the gut microbiota profiling showed altered levels of several other microbes. Their potential role has not been discussed in the manuscript.

Response:

We appreciate the reviewer's suggestion. In addition to *Escherichia-Shigella* and *Klebsiella*, our gut microbiota profiling identified several other microbes, including *Lactococcus*, *Clavibacter*, and *Leuconostoc*. These microbes exhibited significant increases in volunteers who experienced severe AMS. However, their average relative abundances were low, with *Lactococcus* at 0.0036%, *Clavibacter* at 0.0026%, and *Leuconostoc* at 0.0026%. Therefore, we hypothesize that these microbes only play a negligible role in high-altitude pulmonary edema. This information has been incorporated into the discussion section of the revised manuscript and is highlighted in yellow in the marked-up manuscript (line numbers: 606-611).

3. I do not agree with the response of the authors to Q8 raised by the reviewer. The method indicated is not an appropriate way of determining sample size for a gut microbiota study.

Response:

We appreciate the reviewer's suggestion. As the reviewers do not support the use of the species accumulation curve for assessing sample size sufficiency and have not provided a specific alternative method, we have opted to utilize the Pan/Core species curve to address this issue in the revised manuscript. The results indicate that the curves begin to plateau as the number of samples increases, confirming the adequacy of the sample size (Fig. 1A, B). Furthermore, we note that numerous studies

(References 1-8) also employ the Pan/Core species curve to evaluate sample size adequacy. We believe that the Pan/Core species curve is currently the appropriate method for assessing sample size adequacy.

Reference:

1. Yang, F., Ni, B., Liu, Q., He, F., Li, L., Zhong, X., Zheng, X., Lu, J., Chen, X., Lin, H., Xu, R., He, Y., Zhang, Q., Zou, X., & Chen, W. (2022). Human umbilical cord-derived mesenchymal stem cells ameliorate experimental colitis by normalizing the gut microbiota. *Stem cell research & therapy*, 13(1), 475. <https://doi.org/10.1186/s13287-022-03118-1>
2. Yan, Q., Chen, Y., Gao, E. B., Lu, Y., Wu, J., & Qiu, H. (2025). The characteristics of intestinal microflora in infants with rotavirus enteritis, changes in microflora before and after treatment and their clinical values. *Scientific reports*, 15(1), 4312. <https://doi.org/10.1038/s41598-025-88312-w>
3. Wang, J., Chen, G., Chen, H., Chen, J., Su, Q., & Zhuang, W. (2023). Exploring the characteristics of gut microbiome in patients of Southern Fujian with hypocitraturia urolithiasis and constructing clinical diagnostic models. *International urology and nephrology*, 55(8), 1917–1929. <https://doi.org/10.1007/s11255-023-03662-6>
4. Wang, Y., Zhang, Y., Qian, Y., Xie, Y. H., Jiang, S. S., Kang, Z. R., Chen, Y. X., Chen, Z. F., & Fang, J. Y. (2021). Alterations in the oral and gut microbiome of colorectal cancer patients and association with host clinical factors. *International journal of cancer*, 10.1002/ijc.33596. Advance online publication. <https://doi.org/10.1002/ijc.33596>
5. Zhang, P., Li, J., Miao, Y., Zhao, X., Zhu, L., Yao, J., Wan, M., & Tang, W. (2024). Sheng-Jiang powder ameliorates NAFLD via regulating intestinal microbiota in mice. *Frontiers in microbiology*, 15, 1387401. <https://doi.org/10.3389/fmicb.2024.1387401>
6. Xu, Y., Zhu, B. W., Li, X., Li, Y. F., Ye, X. M., & Hu, J. N. (2022). Glycogen-based pH and redox sensitive nanoparticles with ginsenoside Rh2 for effective treatment of ulcerative colitis. *Biomaterials*, 280, 121077. <https://doi.org/10.1016/j.biomaterials.2021.121077>
7. Liu, Y., Yang, C., Meng, Y., Dang, Y., & Yang, L. (2023). Ketogenic diet ameliorates attention deficit hyperactivity disorder in rats via regulating gut microbiota. *PloS one*, 18(8), e0289133. <https://doi.org/10.1371/journal.pone.0289133>
8. Chen, Y., Ye, S., Shi, J., Wang, H., Deng, G., Wang, G., Wang, S., Yuan, Q., Yang, L., &

Mou, T. (2024). Functional evaluation of pure natural edible Ferment: protective function on ulcerative colitis. *Frontiers in microbiology*, 15, 1367630. <https://doi.org/10.3389/fmicb.2024.1367630>

4. The pipeline (QIIME1) and database used for analysis (Silva128) are quite outdated and more recent versions with better functionality are available. It is currently unclear if this affects the study findings.

Response:

We appreciate the reviewer's comments. We have utilized alternative software and the updated version of the database (Silva138) for our analysis. Details of these changes are provided in the methods section of the revised manuscript (line numbers: 144-148). After increasing the sample size and utilizing new software and databases for microbiome analysis, we found no changes in the major differential species, including *Klebsiella*, *Escherichia-Shigella*, *Lactobacillus*, and *Bifidobacterium* (line numbers: 451-461).

Animal experiments:

1. Given that the isolates are human derived, how are the authors sure that the strains colonized the animal gut? Given that colonization is largely influenced by the host and its diet, human-derived strains may not have colonized the animal at all. Furthermore, administration though the oral route may affect delivery to the gut. Thus, gut microbiota data for animal models after inoculation and treatments may provide better clarity.

Response:

This is an important question. We have included gut microbiota data from animal models following inoculation and treatment in the results section, specifically under the subheading “*E. coli* and *K. pneumoniae* contribute to HAPE under hypoxic conditions.” The results indicate that, after six days of *K. pneumoniae* and *E. coli* transplantation, the relative abundance of *Klebsiella* and *Escherichia-Shigella* was significantly higher than in both the

normoxia control (NC) and hypoxia control (HC) groups (Fig. 2A, B). However, synbiotic treatment led to a significant decrease in the relative abundance of *Klebsiella* and *Escherichia-Shigella*, while the abundance of *Bifidobacterium* and *Lactiplantibacillus* significantly increased (Fig. 2C). These results suggest that the strains did indeed colonize the animal gut (line numbers: 485-491). Additionally, we collected feces from these rats after the transplantation of target strains, diluted the fresh feces with sterile PBS to create suspensions, and cultured them on LB or BBL solid media (Figure R1). After a period of cultivation, some colonies were randomly selected for further purification and molecular identification. We detected the presence of viable target strains in the fresh feces of the rats, providing further evidence that the strains colonized the animal gut.

A

B

Figure R1. Rat fecal suspension was cultured with the corresponding medium. (A) Fecal suspension from rats after the transplantation of *K. pneumoniae* and *E. coli* for 3 days was cultured with LB medium. (B) Fecal suspension from rats after the transplantation of *Bifidobacterium* and *Lactiplantibacillus* for 3 days was anaerobically cultured with BBL medium.

A representative 16S rRNA gene sequence isolated from rats after the transplantation of *K. pneumoniae* and *E. coli* is as follows:

```
GCTACACATGCACTCGAGCGGTAGCACAGAGAGCTTGCTCTCGGGTGACGAGCGGCGGACGGGTGAGTAATGTCTGGGAACTGCCTGATGGAGGGGATAACTACTGGAAACGGTAGCTA
ATACCGCATAACGTCGCAAGACCAAAGTGGGGGACCTTCGGGCCTCATGCCATCAGATGTGCCAGATGGGATTAGCTAGTAGGTGGGGTAACGGCTCACCTAGGCGACGATCCCTAGCTGGT
CTGAGAGGATGACCAGCCACACTGGAAGTCTGAGACAGGTCACGACTCTACGGGAGGCAGCAGTGGGGAATATTGCACAATGGGCGCAAGCCTGATGCAGCCATGCCCGTGTGTGAAGAA
GGCCTTCGGGTTGTAAGCACTTTCAGCGGGGAGGAAGGCGTTAAGTTAATAAATCTTGGCGATTGACGTTACCCGCAAGAAGCACCGGTAACCTCCGTGCCAGCAGCCGGTAATACG
GAGGGTGCAAGCGTTAATCGGAATTAAGTGGCGTAAAGCGCACGACGGCGTCTGTCAAGTCGGATGTGAAATCCCGGGCTCAACCTGGGAACTGCATTGAAACTGGCAGGCTAGAGTCT
GTAGAGGGGGGTAGAATCCAGGTGTAGCGGTGAAATGCGTAGAGATCTGGAGGAATACCGGTGGCGAAGGCGGCCCTGGACAAAGACTGACGCTCAGGTGCGAAAGCGTGGGAGC
AAACAGGATTAGATACCCTGGTAGTCCACGCCGTAACGATGTCGATTTGGAGGTTGTGCCCTTGAGGCGTGGCTTCGGGAGCTAACCGGTTAAATCGACCGCTGGGGAGTACGGCCGAA
GGTAAACTCAAATGAATTGACGGGGCCCGCACAAAGCGGTGGAGCATGTGGTTAATTCGATGCAACGGAAGAACCCTACCTGGTCTTGACATCCACAGAACTIAGCAGAGATGGTTTG
GTGCCTTCGGAACTGTGAGACAGGTGCTGCATGGCTGTCGTCAGCTCGTGTGAAATGTTGGGTTAAGTCCCGCAACGAGCGCAACCCCTATCTTGTGCCAGCGGTTCCGGCCGGA
ACTCAAAGGAGACTGCCAGTGATAAAGTGGAGGAAGGTGGGATGACGTCAGTCAATGCGCCCTACGACCAGGGGTACACACGTGCTACAATGGCATATACAAGAGAAGCGACCTCCG
GAGAGCAAGCGGACCTCATAAAGTATGTCGTAGTCCGGATTGGAGTCTGCAACTCGACTCCATGAAGTGGAAATCGCTAGTAATCGTAGATCAGAATGCTACGGTGAATACGTTCCCGGCCT
TGTACACACCGCCGTCACACCATGGGAGTGGGTTGCAAAAGAAGTAGGTAGCTTAACCTTCGGGAGGGCGCTTACCCT
```

A representative 16S rRNA gene sequence isolated from rats after the transplantation of *Bifidobacterium* and *Lactiplantibacillus* is as follows:

```
AGACGGCTCCATCCACAAGAGTTAGGCCACCGGCTTCGGGTGCTGCCACTTTCATGACTTGACGGGCGGTGTGTACAAGGCCGGGAACGCATTACCCGACGTTGCTGATTCGCGAT
TACTAGCGACTCCGCCTTACGACAGTCGAGTTGCAGACTGCGATCCGAACTGAGACCGGTTTTTCAGGGATCCGCTCCGCGTCCCGCTCGCATCCCGTTGACCGGCCAATTGATGATGCGT
GAAGCCCTGGAGTAAGGGGATGATGATCTGACGTCATCCCCACCTTCTCCGAGTTAACCCCGCGGTCCCGTGTAGTCCCGGATAATCCGCTGGCAACACGGGGGAGGGTTGCGC
TCGTTGCGGACTTAACCAACATCTCACGACACGAGCTGACGACGACCATGCACCACCTGTGAACCCGCCCGAAGGGAAAGCCGTATCTTACGACCGTCCGGAAACATGTCAAGCCAGGT
AAGGTTCTTCGCGTGCATCGAATAATCCGATGCTCCCGGCTGTGCGGGCCCCGTCATTTCTTTGAGTTTTAGCCTTGCGGCGTACTCCCAGCGGGATGCTTAACCGCTTAGCTCC
GACACGGAACCCGTGGAAACGGGCCCCACATCCAGCATCCACCGTTTACGGCGTGGACTACCAGGGTATCTAATCTGTTCGCTCCCAACGCTTTCGCTCCTCAGCGTCAGTAACGGCCAGAG
ACCTGCCTTCGCCAATTGTTCTTCCCGATATCTACACATCCACCGTTACACCGGAATTCCAGTCTCCCTACCGCACTCAAGCCCGCCGTACCCGGCGGGATCCACCGTTAAGCGATG
GACTTTCACACCGGACGCGACGAACCGCTACGAGCCCTTACGCCAATAATTCCGGATAACGCTTGCACCCTACGTATTACCGCGGCTGCTGCACGATGTTAGCCGGTCTTATTAACGG
GTAAACTCACTCTCGCTGCTCCCGATAAAAGAGGTTTACAACCGAAGGCTCCATCCCTACGCGGCGTGCATCAGGCTTGCGCCAATTGTCAATATCCCCTGCTGCTCCCG
TAGGAGTCTGGGCGTATCTCAGTCCCAATGTGGCGGTCGCCCTCTCAGGCGGCTACCCGTGAAAGCCACGGTGGGCGTTACCCCGCGTCAAGCTGATAGGACGCGACCCATCCCAT
CCGCGAAAGCTTCCAGAAAGACCATGCGATCAACTGGAACATCCGGCATTACCACCGTTTCCAGGAGCTATTCCGGTGTATGGGGCAGGTCGGTACGCACTTACTACCCGTTCCGCACTC
TCACCACCAAGCAAGCTTGATGGATCCCGTTCGACTGCATGTGAA
```

2. Have the authors considered that the *E. coli* and *K. pneumoniae* used in their study could be pathogenic and may trigger a systemic response? Also, *K. pneumoniae* can

easily disseminate from the site of inoculation to distal body sites, triggering responses there.

Response:

We sincerely appreciate the reviewers for their valuable suggestions and share their concerns. We have included relevant experimental results in the discussion section (line numbers: 629-633). After the transplantation of *K. pneumoniae* and *E. coli* in rats, we cultured 100 μ L of rat blood and lung tissue homogenates on basal salt medium containing 1% glucose, and no culturable colonies were observed (Fig. S9). These results confirm that the administration of *E. coli* and *K. pneumoniae* does not trigger a systemic response or lead to the spread of these bacteria from the site of inoculation to the lungs.

3. The methods do not mention the actual doses of the microbes delivered. The dosage and the rationale for selection of dosage needs to be indicated in the manuscript.

Response:

We appreciate the reviewer's comments. We have rewritten this information regarding the strain concentrations for gavage under the subheading "Isolation, Identification, and Preparation of Strains" in the MATERIALS AND METHODS section of the revised manuscript, which is highlighted in yellow in the Marked-Up Manuscript (line numbers: 174-178). Each rat received a dose of 10 mL/kg by gavage, as detailed under the subheading "Animal Experiment" in the MATERIALS AND METHODS section (line numbers: 203-204). Additionally, we have included references in the revised manuscript to clarify the rationale for the chosen dose (line numbers: 174-178). While various studies report different concentrations of strains for gavage, they typically range from 1×10^8 cfu/mL to 1×10^{10} cfu/mL (References 1-5). In this study, we selected a higher concentration of 5×10^9 cfu/mL (References 6,7) to ensure an adequate number of viable bacteria for colonization in the rat gut, thereby promoting functional activity.

Reference:

1. Wang, D., Zhang, T., Qiu, L., & Zhao, C. (2024). The Potential of the Probiotic Isolate *Lactobacillus plantarum* SS18-50 to Prevent Colitis in Mice. *Food science &*

nutrition, 13(1), e4657. <https://doi.org/10.1002/fsn3.4657>

2. Zeng, Z., Yue, W., Kined, C., Raciheon, B., Liu, J., & Chen, X. (2023). Effect of *Lysinibacillus* isolated from environment on probiotic properties and gut microbiota in mice. *Ecotoxicology and environmental safety*, 258, 114952. <https://doi.org/10.1016/j.ecoenv.2023.114952>

3. Park, B. H., Kim, I. S., Park, J. K., Zhi, Z., Lee, H. M., Kwon, O. W., & Lee, B. C. (2021). Probiotic effect of *Lactococcus lactis* subsp. *cremoris* RPG-HL-0136 on intestinal mucosal immunity in mice. *Applied Biological Chemistry*, 64(1), 93. <https://doi.org/10.1186/s13765-021-00667-6>

4. Bamgbose, T., Quadri, A., Abdullahi, I. O., Inabo, H. I., Bello, M., Kori, L. D., Anvikar, A. R., de la Fuente, J., Piloto-Sardiñas, E., & Cabezas-Cruz, A. (2024). Antiplasmodial Activity of Probiotic *Limosilactobacillus fermentum* YZ01 in *Plasmodium berghei* ANKA Infected BALB/c Mice. *Journal of tropical medicine*, 2024, 6697859. <https://doi.org/10.1155/jotm/6697859>

5. Yang, S., Qiao, J., Zhang, M., Kwok, L. Y., Matijašić, B. B., Zhang, H., & Zhang, W. (2024). Prevention and treatment of antibiotics-associated adverse effects through the use of probiotics: A review. *Journal of advanced research*, S2090-1232(24)00230-3. Advance online publication. <https://doi.org/10.1016/j.jare.2024.06.006>

6. Ishii, T., Furuoka, H., Kaya, M., & Kuhara, T. (2021). Oral Administration of Probiotic *Bifidobacterium breve* Improves Facilitation of Hippocampal Memory Extinction via Restoration of Aberrant Higher Induction of Neuropilin in an MPTP-Induced Mouse Model of Parkinson's Disease. *Biomedicines*, 9(2), 167. <https://doi.org/10.3390/biomedicines9020167>

7. Guo, W., Mao, B., Cui, S., Tang, X., Zhang, Q., Zhao, J., & Zhang, H. (2022). Protective Effects of a Novel Probiotic *Bifidobacterium pseudolongum* on the Intestinal Barrier of Colitis Mice via Modulating the Ppar γ /STAT3 Pathway and Intestinal Microbiota. *Foods (Basel, Switzerland)*, 11(11), 1551. <https://doi.org/10.3390/foods11111551>

4. As per the methods, 20 ml of the synbiotic (~10⁹ CFU/ml) was administered daily for 3 days. Again, what was the rationale for selection of this dosage.

Response:

Thank you for your comments. We would like to clarify that we did not specify in the methods section that 20 mL of the synbiotic (~10⁹ CFU/ml) was administered daily for 3 days. Instead, we stated that each rat received a gavage dose of 10 mL/kg body weight (line numbers: 203-204). For a rat weighing 260 g, this corresponds to a gavage dose of 2.6 mL. This dosage is commonly used in animal studies (References 1-3).

Reference:

1. Zou, Y., Ro, K. S., Jiang, C., Yin, D., Zhao, L., Zhang, D., Du, L., & Xie, J. (2024). The anti-hyperuricemic and gut microbiota regulatory effects of a novel purine assimilatory strain, *Lactiplantibacillus plantarum* X7022. *European journal of nutrition*, 63(3), 697–711. <https://doi.org/10.1007/s00394-023-03291-w>
2. Manaer, T., Sailike, J., Sun, X., Yeerjiang, B., & Nabi, X. (2025). Therapeutic effects of composite probiotics derived from fermented camel milk on metabolic dysregulation and intestinal barrier integrity in type 2 diabetes rats. *Frontiers in pharmacology*, 15, 1520158. <https://doi.org/10.3389/fphar.2024.1520158>
3. Song, Y., Sun, M., Ma, F., Xu, D., Mu, G., Jiao, Y., Yu, P., & Tuo, Y. (2024). *Lactiplantibacillus plantarum* DLPT4 Protects Against Cyclophosphamide-Induced Immunosuppression in Mice by Regulating Immune Response and Intestinal Flora. *Probiotics and antimicrobial proteins*, 16(2), 321–333. <https://doi.org/10.1007/s12602-022-10015-9>

5. The control suggested by Reviewer 1 in Q5 (NC + KE control) is critical and the authors have not implemented that same.

Response:

We sincerely appreciate the reviewers for their valuable suggestions. We have incorporated the NC + KE control group into the revised manuscript. The results indicated that there was no HAPE in the rats of the NKE group compared to the NC group. Additionally, the rats in the HKE group exhibited significantly higher indicators related to HAPE compared to the NKE group (Fig. 2G-I; line numbers: 185-186, 497-501). These findings suggest that K.

pneumoniae and *E. coli* exhibit pathogenic effects only at high altitudes, whereas they do not demonstrate harmful effects at low altitudes.

6. In Fig. 2D, the difference is very marginal and there is no need to have a cut graph to highlight the marginal difference.

Response:

We sincerely appreciate the reviewers for their valuable suggestions. This problem has been corrected in the revised manuscript (Fig. 2G).

Cell culture experiments:

1. The query raised by Q4 by Reviewer 1 also remains unanswered. It is unlikely that LPCs can accumulate at tested (of 25 µg/mL 100 µg/mL) concentration ranges under physiological conditions. The cited manuscript indicates that LPC levels are higher in AMS-sensitive group, but does not indicate concentrations. This issue is still unresolved.

Response:

We sincerely appreciate the reviewers for their valuable feedback. We apologize for not quantifying LPC levels in the plasma of AMS patients. The primary reasons for this omission are as follows: first, due to technical limitations, we are unable to transport blood samples from AMS patients to another district for further analysis. Second, LPCs represent a class of chemical constituents with numerous distinct structures, complicating their quantitative analysis; thus, considerable systematic research is still required. Additionally, we were unable to locate relevant literature quantifying LPC levels in patients with AMS or HAPE, indicating that few previous studies have explored the relationship between LPCs and HAPE.

Other minor comments:

1. The manuscript was submitted without line numbers, which made it difficult to review.

Response:

We sincerely appreciate the reviewers for their valuable feedback. We understand that submitting the manuscript without line numbers created some inconvenience during the

review process, and we apologize for this oversight. We will ensure that line numbers are included in all future submissions.

2. The introduction mentions shotgun metagenomics, while the analysis carried out was 16S-based microbiota profiling.

Response:

We thank the reviewer for highlighting this mistake, which has been corrected in the revised manuscript. The correction is marked in yellow in the Marked-Up Manuscript (line numbers: 101).

3. Figure legends can still be improved to include more details on the experimentation and statistics, which will make them self-sufficient.

Response:

We sincerely appreciate the reviewers for their valuable feedback. We have made the necessary revisions to the figure legends in the revised manuscript. These changes are highlighted in yellow in the marked-up manuscript (line numbers: 901-982).

Reviewer #2 (Comments for the Author):

The article overall is a comprehensive study, but it has some potential limitations.

1. Firstly, the sample size of volunteers in each group is relatively small (only 15 in some groups), which may limit the statistical power and generalizability of the results.

Response:

We sincerely appreciate the reviewers for their valuable feedback. In the revised manuscript, we have increased the sample size from 15 to 30 participants in each group. Subsequently, we assessed the adequacy of the sample size for each group using the Pan/Core species curve. The results indicated that the curves began to plateau as the number of samples increased, confirming that the sample size is adequate for this study (Fig. 1A, B; line numbers: 437-439). Furthermore, we noted that many studies involving human subjects also reported sample sizes around 30 participants per group (References 1-8).

Reference:

1. Iadsee, N., Chuaypen, N., Techawiwattanaboon, T., Jinato, T., Patcharatrakul, T., Malakorn, S., Petchlorlian, A., Praditpornsilpa, K., & Patarakul, K. (2023). Identification of a novel gut microbiota signature associated with colorectal cancer in Thai population. *Scientific reports*, 13(1), 6702. <https://doi.org/10.1038/s41598-023-33794-9>
2. Lazarevic, V., Teta, D., Pruijm, M., Stoermann, C., Marangon, N., Mareschal, J., Solano, R., Wurznier-Ghajarzadeh, A., Gaia, N., Cani, P. D., Dizdar, O. S., Herrmann, F. R., Schrenzel, J., & Genton, L. (2024). Gut microbiota associations with chronic kidney disease: insights into nutritional and inflammatory parameters. *Frontiers in microbiology*, 15, 1298432. <https://doi.org/10.3389/fmicb.2024.1298432>
3. Guo, Y. P., Shao, L., Wang, L., Chen, M. Y., Zhang, W., & Huang, W. H. (2021). Bioconversion variation of ginsenoside CK mediated by human gut microbiota from healthy volunteers and colorectal cancer patients. *Chinese medicine*, 16(1), 28. <https://doi.org/10.1186/s13020-021-00436-z>
4. Wang, Q., Chen, C., Zuo, S., Cao, K., & Li, H. (2023). Integrative analysis of the gut microbiota and faecal and serum short-chain fatty acids and tryptophan metabolites in patients with cirrhosis and hepatic encephalopathy. *Journal of translational medicine*, 21(1), 395. <https://doi.org/10.1186/s12967-023-04262-9>
5. Luangphiphat, W., Prombutara, P., Muangsillapasart, V., Sukitpunyaroj, D., Eeckhout, E., & Taweechotipatr, M. (2024). Exploring of gut microbiota features in dyslipidemia and chronic coronary syndrome patients undergoing coronary angiography. *Frontiers in microbiology*, 15, 1384146. <https://doi.org/10.3389/fmicb.2024.1384146>
6. Xia, M., Xu, Y., Li, H., Huang, J., Zhou, H., Gao, C., & Han, J. (2024). Structural and functional alteration of the gut microbiota in elderly patients with hyperlipidemia. *Frontiers in cellular and infection microbiology*, 14, 1333145. <https://doi.org/10.3389/fcimb.2024.1333145>
7. Lee, P. C., Wu, C. J., Hung, Y. W., Lee, C. J., Chi, C. T., Lee, I. C., Yu-Lun, K., Chou, S. H., Luo, J. C., Hou, M. C., & Huang, Y. H. (2022). Gut microbiota and metabolites associate with outcomes of immune checkpoint inhibitor-treated unresectable hepatocellular carcinoma. *Journal for immunotherapy of cancer*, 10(6), e004779. <https://doi.org/10.1136/jitc-2022-004779>
8. Khiabani, S. A., Haghghat, S., Khosroshahi, H. T., Asgharzadeh, M., & Kafil, H. S. (2022).

Clostridium species diversity in gut microbiota of patients with renal failure. Microbial pathogenesis, 169, 105667. <https://doi.org/10.1016/j.micpath.2022.105667>

2. Secondly, the animal models used may not fully mimic the complex pathophysiology of HAPE in humans.

Response:

We sincerely appreciate the reviewers for their valuable feedback. Due to significant differences between the gut microbiota of experimental rats and humans, the rat model may not fully replicate HAPE in humans. However, it is evident that opportunistic pathogens, such as *Escherichia coli* and *Klebsiella pneumoniae*, play a role in the promotion of HAPE. Our findings provide valuable insights into the pathophysiology of HAPE in humans.

3. Thirdly, although the study proposes a gut microbiota - LPCs/inflammation - HAPE axis, there may be other factors or pathways not explored that could also be involved in the development of HAPE.

Response:

We sincerely appreciate the reviewer's insights and agree with their perspective. Current literature predominantly supports diffuse hypoxic pulmonary vasoconstriction as a pathophysiological basis for HAPE, alongside other contributing factors such as sympathetic stimulation, reduced nitric oxide bioavailability, increased endothelin levels, and decreased reabsorption of interstitial fluid, all of which collectively facilitate the development of HAPE (References 1,2). In this study, we introduce a novel perspective suggesting that gut opportunistic pathogens contribute to HAPE by elevating lysophosphatidylcholines and inducing inflammation. However, the inability of gut pathogens to trigger HAPE at low altitudes indicates that their harmful effects must occur in conjunction with other factors. These factors may include increased pulmonary microvascular pressure due to high-altitude exposure, hypoxia-induced damage to the intestinal barrier that allows significant quantities of harmful substances to enter systemic circulation from the gut, and decreased reabsorption of alveolar interstitial fluid. In light of existing reports and our understanding of HAPE, it remains essential to conduct comprehensive research to elucidate the influence of these factors on HAPE.

Reference:

1. Sharma Kandel, R., Mishra, R., Gautam, J., Alaref, A., Hassan, A., & Jahan, N. (2020). Patchy

Vasoconstriction Versus Inflammation: A Debate in the Pathogenesis of High Altitude Pulmonary Edema. *Cureus*, 12(9), e10371. <https://doi.org/10.7759/cureus.10371>

2. Richalet, J. P., Jeny, F., Callard, P., & Bernaudin, J. F. (2023). High-altitude pulmonary edema: the intercellular network hypothesis. *American journal of physiology. Lung cellular and molecular physiology*, 325(2), L155–L173. <https://doi.org/10.1152/ajplung.00292.2022>

We have made every effort to enhance the manuscript and have implemented several revisions. These changes do not affect the key results or the overall framework of the paper. We have highlighted these modifications in yellow in the marked-up manuscript. We sincerely appreciate the warm and professional efforts of the Editors and Reviewers, and we hope that these corrections will receive your approval. Once again, we thank you very much for your comments and suggestions.

Re: Spectrum03057-24R1 (**Gut opportunistic pathogens contribute to high altitude pulmonary edema by elevating lysophosphatidylcholines and inducing inflammation**)

Dear Prof. jingming Jia:

Your manuscript has been accepted, and I am forwarding it to the ASM production staff for publication. Your paper will first be checked to make sure all elements meet the technical requirements. ASM staff will contact you if anything needs to be revised before copyediting and production can begin. Otherwise, you will be notified when your proofs are ready to be viewed.

Sincerely,
Zhe LYU
Editor
Microbiology Spectrum